# Genetic isolation and metabolic complexity of an Antarctic subglacial microbiome

Kyung Mo Kim [1,6] ✉, Kyuin Hwang [1,6], Hanbyul Lee [1], Ahnna Cho [1], Christina L. Davis[2,5], Brent C. Christner [2], John C. Priscu [3,4] ✉ & Ok-Sun Kim [1] ✉

Microbes inhabiting and evolving in aquatic ecosystems beneath polar ice sheets subsist under energy-limited conditions while in relative isolation from surface gene pools and their common ancestral populations of origin. Samples obtained from beneath West Antarctic Ice Sheet (WAIS) allowed us to examine evolutionary relationships of and identify metabolic pathways in microbial genomes recovered from the Mercer Subglacial Lake (SLM) ecosystem. We obtained 1,374 single-cell amplified genomes (SAGs) from individual bacterial and archaeal cells that were isolated from samples of SLM's water column and sediments. These genomes reveal that a diversity of microorganisms including Patescibacteria exists in SLM. Comparative analyses show that most genomes correspond to new species and taxonomic groups, with phylogenomic and functional evidence supporting their genetic isolation from marine and surface biomes. Genomic data reveal diverse metabolisms in SLM that are capable of oxidizing organic and inorganic compounds via aerobic or anaerobic respiration. Distinct metabolic guild structures are observed for the subglacial populations, where trophic shifts from organotrophy to chemolithotrophy may depend on oxygen availability. Our SAG data suggest versatile metabolic capabilities in the characterized microbial assemblage, reveal key energy-generating strategies in the subglacial aquatic ecosystem, and provide a framework to assess microbial evolution beneath WAIS.

Ice streams along West Antarctica's Siple Coast flow over deformable sediments saturated with water[1], and overlie an extensive subglacial hydrological network that exists beneath the ice[2]. Sediments upstream from the modern West Antarctic Ice Sheet (WAIS) grounding line consist of crustal sedimentary infill and overridden marine sediment, indicating the ice sheet extent has varied over geological time[3–6]. The subglacial hydrological systems beneath the Whillans Ice Stream (WIS) and Mercer Ice Stream (MIS) consist of numerous lakes connected by a network of drainage pathways, with episodic flows eventually discharging water[7], solutes[8], and sediments[9] to the marine cavity beneath Ross Ice Shelf[10]. Research in this region has revealed the roles of water and sediments in ice stream flow[11], the historical dynamics of WAIS[4–6,12–14], and the geobiology of its subglacial aquatic ecosystems[15–19]. Given that genomic[13] and glaciological[14] evidence implies the form, evolution, distribution, and function of subglacial ecosystems were heavily influenced by ice sheet extent and sea level from the Late Pleistocene to Middle Holocene[12], new observations from beneath WAIS provide a means to evaluate these hypotheses.

[1]Division of Life Sciences, Korea Polar Research Institute, 26 Songdomirae-ro, Yeonsu-gu, Incheon 21990, Republic of Korea. [2]Department of Microbiology and Cell Science, University of Florida, Gainesville, FL, USA. [3]Department of Land Resources and Environmental Sciences-Emeritus, Montana State University, Bozeman, MT, USA. [4]Desert Research Institute, Reno, NV, USA. [5]Present address: Department of Natural Resource Sciences, McGill University, Ste-Anne-de-Bellevue, QC, Canada. [6]These authors contributed equally: Kyung Mo Kim, Kyuin Hwang. ✉e-mail: kmkim@kopri.re.kr; jpriscu@montana.edu; oskim@kopri.re.kr

Drilling projects that explored subglacial lakes in West Antarctica [i.e., Whillans Subglacial Lake (SLW) and Mercer Subglacial Lake (SLM)] have discovered metabolically active microbial communities[12,15,16,20] and evidence for metacommunity structure across distinct subglacial hydrological basins[17]. Because WAIS is a barrier to mixing between marine, surface, and subglacial systems, the sources of inorganic nutrients and organic matter to the lakes are basal ice melting[21], rock comminution[22], material stored in the sediments[16], and in-situ water column and sedimentary transformations[9,12]. In the absence of sunlight, the metabolic energy sources available in the subglacial environment include the oxidation of methane[19] and reduced nitrogen, iron, and sulfur compounds[15,16,18,20]. The processes driving microbial metabolisms in SLW and SLM extend beyond the subglacial ecosystems, exerting influence on the geochemical and biological systems as they discharge to the Southern Ocean. For instance, several studies have highlighted the potential for nutrients in subglacial discharge to affect oceanic primary productivity[3], bacterial productivity[21], and the level of greenhouse gases[3,19].

SLM is overlain by 1,087 m of ice, is located near the confluence of MIS and WIS, and formed ~180 y before present with stagnation of Kamb Ice Stream[9] (Fig. 1). When sampled in December 2018, SLM had a surface area of ~143 km$^2$, and a water column depth of ~15 m deep at the drill location[23]. Approximately 80% of the freshwater in SLM is sourced from WAIS basal melt, with the remainder originating from East Antarctica[7,12]. The lake undergoes a fill-drain cycle every 4 to 6 years[9]. Measurements in the water column of SLM indicated a temperature of −0.74 °C, a pH of 8.2, low dissolved organic carbon (DOC; ~30 μM), low inorganic nutrients ($NH_4^+$, $NO_3^-$ and soluble reactive P of 0.5, 3.0 and 2.0 μM, respectively), and oxygen supersaturation[17,23,24]. Sediment cores retrieved from SLM consisted of a ~12 cm thick surface unit of finely laminated material deposited from glacially-sourced lake water. The laminated sediments overlie diamict that pre-dates the lake and

were deposited from the upstream catchment[9]. Sediment porewater profiles for chloride and conductivity that increase with depth indicate an influence of past marine incursion events, the most recent of which occurred during the Middle Holocene[12].

We used a single-cell genomic approach to recover microbial genomes from the water column and surficial sediments of SLM. Current information on the structure and function of microbial communities in WAIS subglacial lakes has largely relied on studies of 16S rRNA genes[15–17], metabolic marker genes[18,19], and biogeochemical data[20,21]. Given the limitations of single-gene datasets for inferring evolutionary relationships and metabolic diversity, our single-cell genomic data provides a large complement of identifiable functional genes for the prediction of metabolic pathways and to conduct robust phylogenomic analysis. Single-cell genomics has several advantages over shotgun metagenomics, including the ability to resolve strain heterogeneity within a species that is collapsed within metagenome assemblies[25], as well as being ideal for samples in which recoverable cell biomass is too low to yield sufficient DNA for robust shotgun library construction, sequencing, and assembly[26]. Our data from 1374 single-cell amplified genomes (SAGs) allowed us to reconstruct metabolic pathways contributing to biogeochemical cycling in SLM and offer insight into the emerging understanding of microbial evolution beneath WAIS.

## Results

### Microbial diversity of SLM SAGs

A total of 3170 microbial cells were sorted from four filter-concentrated lake water samples (>3 μm, 3 to 0.8 μm, 0.8 to 0.2 μm, and <0.2 μm), five sediment core samples (10 cm depth, sectioned at 2 cm intervals), and a bulk surficial sediment sample using flow cytometry[27]. After genome amplification and sequencing, bioinformatic analyses identified 1,374 SAGs with <5% CheckM contamination

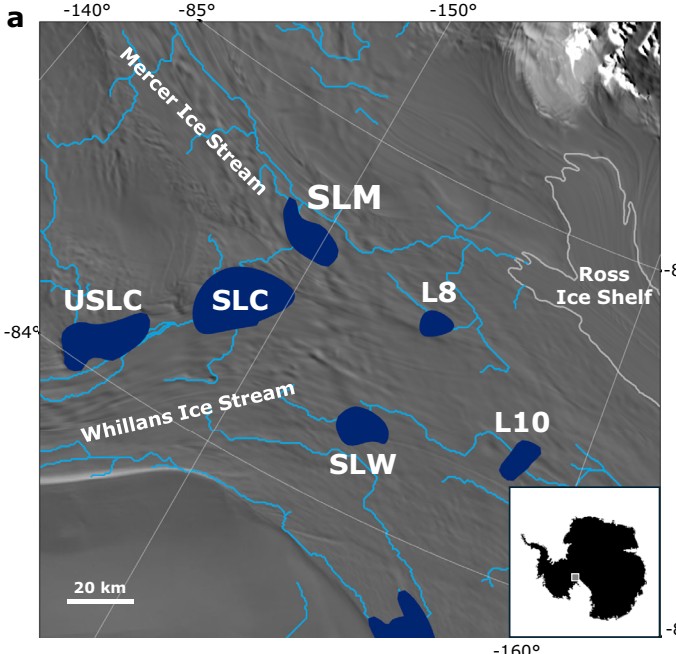

| Physicochemical property | |
| --- | --- |
| Surface area | 143 km$^2$ |
| Depth | 15 m |
| Ice thickness | 1,087 m |
| Temperature | -0.74°C |
| pH | 8.2 |
| DOC (μm) | 30 |
| $NH_4^+$ (μm) | 0.5 |
| $NO_3^-$ (μm) | 3.0 |
| Soluble P (μm) | 2.0 |

**Fig. 1 | Location and physicochemical characteristics of Mercer Subglacial Lake (SLM). a** Geographical setting of SLM beneath West Antarctic Ice Sheet (WAIS), showing its position relative to glacial ice streams and the grounding line. The background satellite image is from the MEaSUREs MODIS Mosaic of Antarctica 2013-2014 (MOA2014) Image Map, Version 1 (NSIDC)[97]. Subglacial lake polygons (navy) are based on the Antarctic Active Subglacial Lake Inventory from ICESat altimetry (NSIDC)[98]. The light blue lines indicate the subglacial water flow paths beneath WAIS[99], and the grounding line (light grey line) is derived from MEaSUREs

Antarctic Grounding Line from Differential Satellite Radar Interferometry, Version 2[100]. Image/photo courtesy of the National Snow and Ice Data Center, University of Colorado, Boulder. Subglacial lake names: Upper Subglacial Lake Conway (USLC), Subglacial Lake Conway (SLC), Subglacial Lake Whillans (SLW), Lake 8 (L8), and Lake 10 (L10). **b** Physicochemical properties of SLM lake water, including lake depth, surface area, overlying ice thickness, basal ice temperature, and concentrations of key nutrients[2,9,17,23,24].

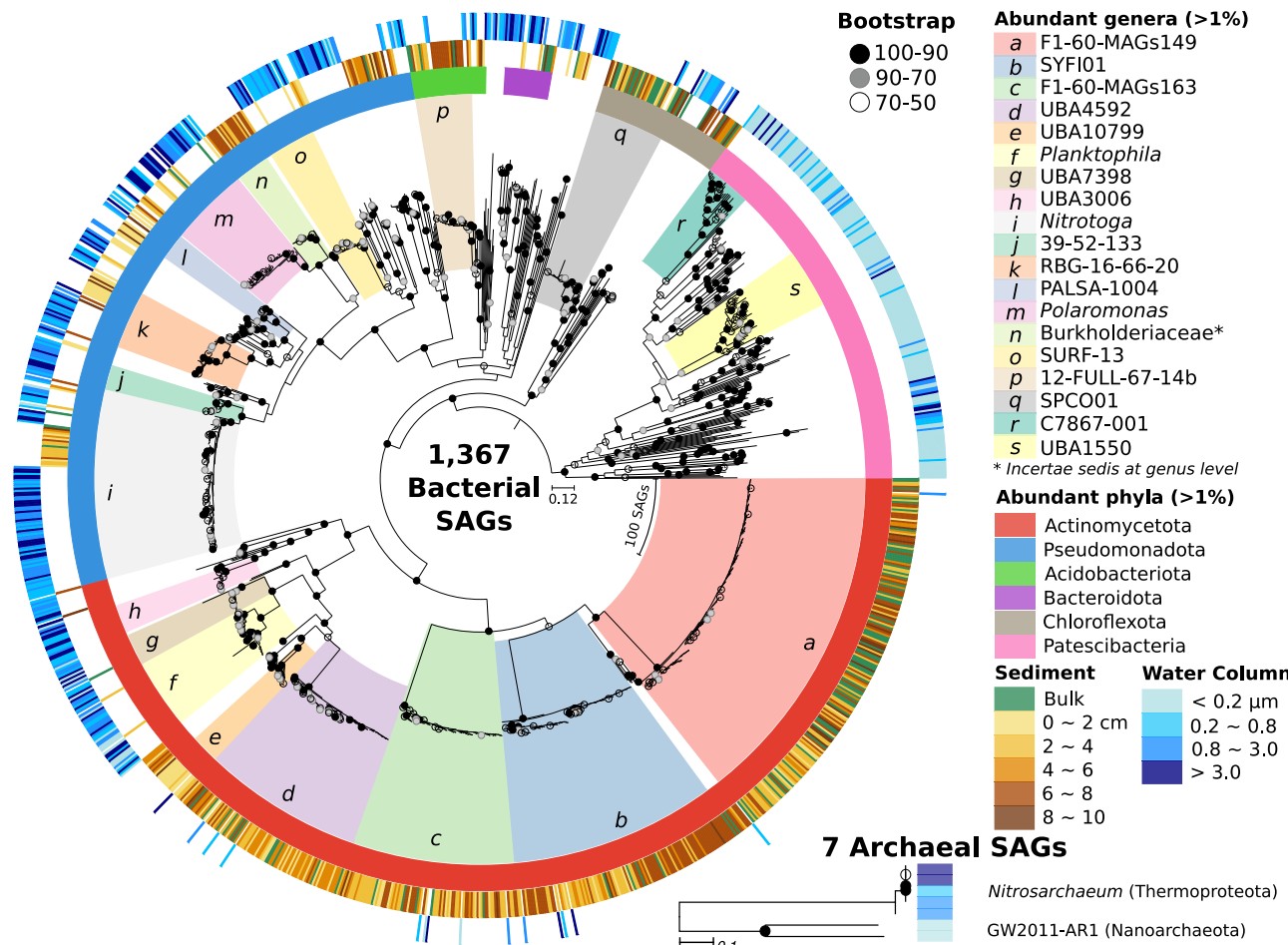

**Fig. 2 | Phylogenomic trees of bacteria (1367 SAGs) and archaea (7 SAGs) of Subglacial Lake Mercer.** Two maximum likelihood trees were reconstructed using RAxML from the concatenated GTDB protein sequence alignment (5035 aligned sites) of 120 bacterial marker genes and from that (10,100 aligned sites) of 53 archaeal marker genes, respectively. The bacterial tree was rooted by the most recent common ancestral node of Patescibacteria. The archaeal tree is unrooted.

From the innermost to the outmost, colors indicate the most abundant genera, phyla, the depth of sediment, and microbial cell sizes of the water column, respectively. Bootstrap support values were indicated on the nodes using black, grey, and open circles. An asterisk indicates the unclassified genus Burkholder-aceae incertae sedis. Source data are provided as a Source Data file.

rate (Supplementary Note 1)[28]. Using GTDB-Tk[29], the SAG sequences were aligned to GTDB marker genes and classified by their taxonomic placement into the GTDB reference tree[30]. Given that the placement-based approach can produce incorrect taxonomic assignments when queries are distantly related to reference genomes[31], the taxonomy inferred from these alignments was evaluated using FastTree[32]. This indicated that 57 of the 1,374 SAGs were misidentified (e.g., Supplementary Fig. 1), so the taxonomic assignment for these SAGs were revised based on FastTree phylogeny (Supplementary Data 1).

Analysis of the curated SAGs revealed differences in the taxonomic composition (Fig. 2) and structure (Supplementary Fig. 2a) of assemblages from the water column and sediment (P-value < 0.05 by PERMANOVA test). The clear taxonomic differences between the microbes in water and sediments suggest that the two habitats are environmentally heterogeneous. The six sediment samples exhibit similar microbial community structures (Supplementary Fig. 2a); hence, they were pooled into a single sample (total of 756 SAGs, hereafter called sediment_SAGs). The three water samples (fractions >0.2 μm) sharing similar community structures were also pooled (a total of 428 SAGs, hereafter called lake_SAGs), while the remaining water sample fraction (<0.2 μm) was examined separately (a total of 190 SAGs, hereafter called lake_small_SAGs).

The sediment_SAGs classify within seven bacterial phyla, but the vast majority are Actinomycetota (69.2% SAGs) and Pseudomonadota

(16.8%) (Fig. 2 and Supplementary Data 2). Similarly, the set of lake_SAGs showed high abundances of Pseudomonadota (53.7%) and Actinomycetota (19.9%) among the 14 bacterial phyla and 1 archaeal phylum identified, while Patescibacteria (83.7% of lake_small_SAGs) and Actinomycetota (10.0%) were the most abundant phyla of the eight comprising the lake_small_SAGs. Archaeal SAGs were rare in the dataset and represented by five lake_SAGs affiliated with the genus *Nitrosarchaeum* and two lake_small_SAGs affiliated with the family GW2011-AR1 (the phylum Nanoarchaeota). Only two SAGs were taxonomically assignable at the species level (Supplementary Data 1). Nearly 95% of the SAGs (1296) could be identified as members of 162 GTDB genera (e.g., *Polaromonas* and UBA1004). Of the remaining 76 SAGs that could not be classified at the lowest levels of the taxonomic hierarchy, 5 belong to unclassified families (e.g., unclassified Hydro-genedentiales), and 71 are members of 28 unclassified genera (e.g., unclassified Nanopelagicaceae). The CHAO1 statistic predicts 351 genera and indicates that the 162 genera (inc. two species) and 33 higher taxonomic groups comprising the 1374 SAGs account for approximately half of the microbial richness in SLM. The rarefaction analysis also demonstrated that the SAG data underrepresented the diversity of the water and sediment communities (Supplementary Fig. 3a).

The SAG data indicated that viruses are integral components of the SLM ecosystem. VirSorter which searches for viral hallmark genes

and viral-like genes detected four phage contigs (4 SAGs) and two prophage-containing contigs (2 SAGs) as the most confident predictions[33]. The less confident prediction of VirSorter yielded 98 phage contigs and 50 prophages from 89 and 49 SAGs, respectively. Except for five phage contigs and three prophages, all of the phage contigs and prophages contain viral hallmark genes and viral-like genes that classify within Caudovirales. In addition, CRISPR spacer sequences were detected in 854 SAGs at an average frequency of 3.2 copies per SAG, providing direct evidence for past viral infection of SLM's microbial populations.

A total of 13,022 amplicon sequence variants (ASVs) from SLM, SLW, and Whillans Grounding Zone (WGZ)[17] were compared to 686 SAGs containing 16S rRNA sequences longer than 1000 bp (Supplementary Data 1). Of these, 668 SAGs (97.4%) had 16S rRNA sequences with ≥99% identity to 803 ASVs, comprising 616 from SLM, 340 from SLW, and 24 from WGZ (Supplementary Fig. 4b). The remaining 18 SAGs did not match any ASVs. Notably, 645 of the 668 SAGs shared ≥99% sequence identity with the 616 ASVs from SLM, indicating that approximately 94% of the SAGs (645 out of 686) are represented in the SLM ASV dataset. Comparison of abundance-weighted beta diversity showed that community structure based on the SAG-derived 16S rRNA sequences was distinct from that based on analysis of ASVs of SLM ($P$-value < 0.05 by PERMANOVA) (Supplementary Fig. 2b, c). The disparity in taxonomic abundance between assemblages in the SAG and ASV datasets is likely attributable to well established technical biases associated with single-cell sorting, whole genome amplification, and 16S rRNA sequencing, including differences based on cell size, lysis efficiency, particle association, taxon abundance, GC content, and primer affinity (Supplementary Note 2)[25].

## Evolutionary divergence of SLM SAGs

Phylogenomic analysis of the 19 most abundant genera (genera with abundance > 1% of the total SAGs; 71% of the 1,374 SAGs; Fig. 2) revealed that most SAGs cluster together and that they are phylogenetically separated from genomes from other environmental sources ($P$-value < 0.05 by genealogical sorting index test[34], Fig. 3a and Supplementary Figs. 6–25). To assess evolutionary divergence, we examined genomic average nucleotide identity (ANI). ANI values for 1,367 SAGs to their closet GTDB genomes average 77.0% (Fig. 3b), while the remaining 7 SAGs, which showed less than 10% alignment with any GTDB reference genome, were classified as 'No match'. ANI values were also calculated for the 64 of 66 SAGs with <5% CheckM contamination that were recovered from seawater beneath Ross Ice Shelf (RIS; EMBL, Bioproject PRJEB35712)[35]. This site is ~800 km from the location where outflow from SLM enters the marine cavity below RIS and its SAGs were more similar to their closet GTDB genomes than the SLM SAGs ($P$ < 0.01; two-tailed $t$-test), having an average ANI of 87.5%. Only one RIS SAG was classified as 'No match'. We also computed the amino acid sequence similarity of individual SAG proteins against all proteins in the NCBI non-redundant database. To achieve this, the 1,399,955 predicted protein sequences from the 1374 SAGs were grouped into 534,964 clusters based on a > 95% sequence similarity cutoff and blasted against the database. The same procedure was conducted for the 63,006 protein-encoding genes (45,895 clusters) of the 64 RIS SAGs. The results produced distinctly different distributions of protein clusters corresponding to each site and that have significant differences in sequence similarity to their nearest neighbor proteins in GenBank ($P$ < 0.001 by $t$-test; Fig. 3c). The lower mean of amino acid similarity for SLM SAGs (67.8%; RIS, 82.1%) is a genomic feature consistent with a higher degree of evolutionary divergence in Antarctic subglacial microbial populations relative to those inhabiting the marine system.

We reconstructed phylogenomic trees for the 19 most abundant genera (>1% SAG abundance for each; Supplementary Figs. 7–25) to identify sister taxa of the SAGs on rooted trees for examining the possible environmental (e.g., freshwater versus marine) and evolutionary origins of SLM's extant populations. The environmental origins of sister taxa to the SLM lineages were determined by extracting the isolation source from NCBI metadata for 1,810 GTDB genomes taxonomically related to the 19 genera (Supplementary Data 3, 4). The majority of source habitats inferred were terrestrial ecosystems that included freshwater and ice (e.g., groundwater, soil, glacial ice, acidmine drainage, and wastewater), while few related genomes were identified from marine environments (Supplementary Figs. 7–25, Supplementary Data 5). When comparing the relative abundance of metabolic pathways (i.e., the proportion of genomes; see below for details) between SLM taxa and their sister taxa within phylogenomic trees of the most abundant genera, we observed clear functional divergence between these two phylogenetic groups (Supplementary Figs. 7–25; Supplementary Data 10).

## Metabolic pathway inferred from SAG data

The annotation of the SAGs produced an average of 1,058 genes and 550 KEGG Orthology (KO) per SAG. Compared with 14,536 KOs predicted by CHAO1, the 11,645 KOs observed from the 1,374 SAGs represented approximately 80% of the expected functional diversity of SLM (Supplementary Fig. 3b). Metabolic potential of individual SAGs was determined by identifying key enzymes and other criteria (Supplementary Data 6–8). The true proportion of genomes with a specific metabolic potential was estimated by correcting for the incompleteness of the SAGs using the formula from Acker et al. (2022)[36]. The 20 SAGs with <10% completeness inflate values for the proportion of genomes, and therefore, were excluded from this analysis. It is noteworthy that these 20 large genomes exhibit both 0.2% of genomic contamination and a clear correlation between genome size and gene number (Supplementary Fig. 26), suggesting that they are genuine and not the result of bias from whole genome amplification. Bacterial strains within a species exhibit distinct functional traits[37], and strain heterogeneity was evident in our statistical analysis (Supplementary Fig. 28 and Supplementary Note 3), where the observed proportion (52.3%) of KEGG orthologs shared by three near-complete SAGs (89.4% completeness on overage) of the same species (>98% ANI values) was significantly lower ($P$ < 0.01; Monte Carlo test) than the expected value (71.6%). Because bacterial strains of the same species can differ significantly in their ecological functions, we examined the metabolic potential for each SAG.

The 1374 SAGs exhibited an average genome completeness of 38.7% and contamination of 0.4% (Supplementary Data 1). Based on the minimum information standards for microbial genomes[38], approximately 73% of these SAGs were classified as low-quality drafts with <50% completeness. To assess whether the low genome completeness biased the estimation of metabolic potential, we statistically compared the genomic proportions of 70 metabolic pathways between high-quality reference genomes and those subjected to artificial genome reduction. We first identified the closest GTDB genomes for 1229 SAGs based on ANI values, excluding SAGs with extremely low completeness, no close GTDB genome, or higher completeness than their GTDB counterpart. From the 1229 GTDB genomes (average completeness of 82.4%), the true proportion of genomes for each metabolic pathway was estimated while correcting for genome incompleteness[36] (Supplementary Data 9). Next, each GTDB genome was artificially reduced to match the genome completeness of its corresponding SAG (see Methods), and the expected proportion of genomes for individual metabolic pathways were calculated from the set of 1229 reduced GTDB genomes. This process was repeated 1000 times, generating distributions of genomic proportions for the metabolic pathways. Comparison of these distributions to the proportions of the unreduced GTDB genomes revealed that only 4 out of 70 metabolic pathways (glycolysis, gluconeogenesis, glyoxylate cycle and hydrogenase) showed significantly lower representation (fold change <1 and

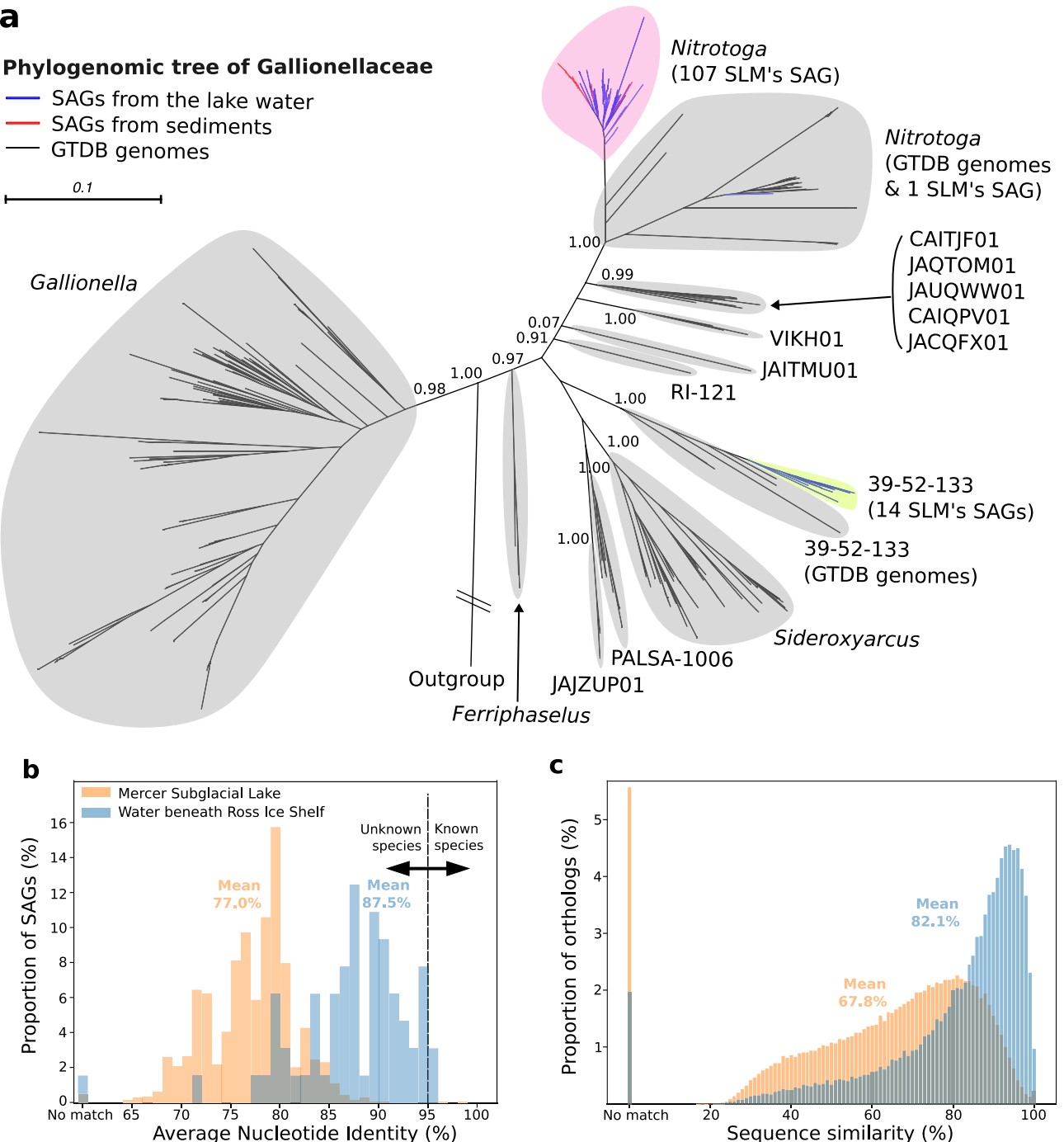

**Fig. 3 | Genetic isolation of SLM's SAGs. a** A phylogenomic tree reconstructed using the concatenated GTDB protein sequence alignment that represents 122 SLM SAGs and 81 GTDB species representative genomes belonging to the family Gallionellaceae. The tree was rooted by an outgroup affiliated with the family SG8-39 of the order Burkholderiales. The 107 out of 108 SLM SAGs belonging to the genus *Nitrotoga* are cohesively clustered (see a circle in pink). The Shimodaira-Hasegawa support values for node confidence are presented next to the branches. **b** A histogram representing genomic average nucleotide identity (ANI) between SAGs and their closest GTDB genomes. The dotted vertical line at 95% sequence identity separates known and unknown bacterial species. When less than 10% of the SAG sequence aligned with any GTDB genomes, the SAG was considered a'No match'. The ANI values for the 1374 SLM SAGs in orange and for 64 SAGs isolated from seawater beneath Ross Ice Shelf (RIS) in blue. **c** Histograms represent sequence similarity between query proteins and BLAST top hits of the NCBI NR database. The query proteins were derived from 537,441 and 45,895 orthologs that were clustered from the 1374 SLM SAGs (distribution in orange) and from the 64 RIS SAGs (distribution in blue), respectively. Two overlapped vertical bars on left represent the number of orthologs without significant BLAST hits under the *E*-value threshold 10. Source data are provided as a Source Data file.

FDR-adjusted $P < 0.05$), suggesting that 94.3% of pathways (66 out of 70) were not significantly underestimated at ~38% genome completeness. However, 19 out of the 70 metabolic pathways were significantly overrepresented (fold change > 1 and FDR-adjusted $P < 0.05$), including the $aa_3$-type cytochrome c oxidase, ED pathway, fermentation metabolisms, nitrogen metabolisms, and sulfur metabolisms (Supplementary Data 9). Thus, caution is warranted when interpreting metabolic potential in these pathways, referring to the observed inflation levels (e.g., fold changes in Supplementary Data 9). Using only higher-quality SAGs (e.g., those with >50% genome completeness) has

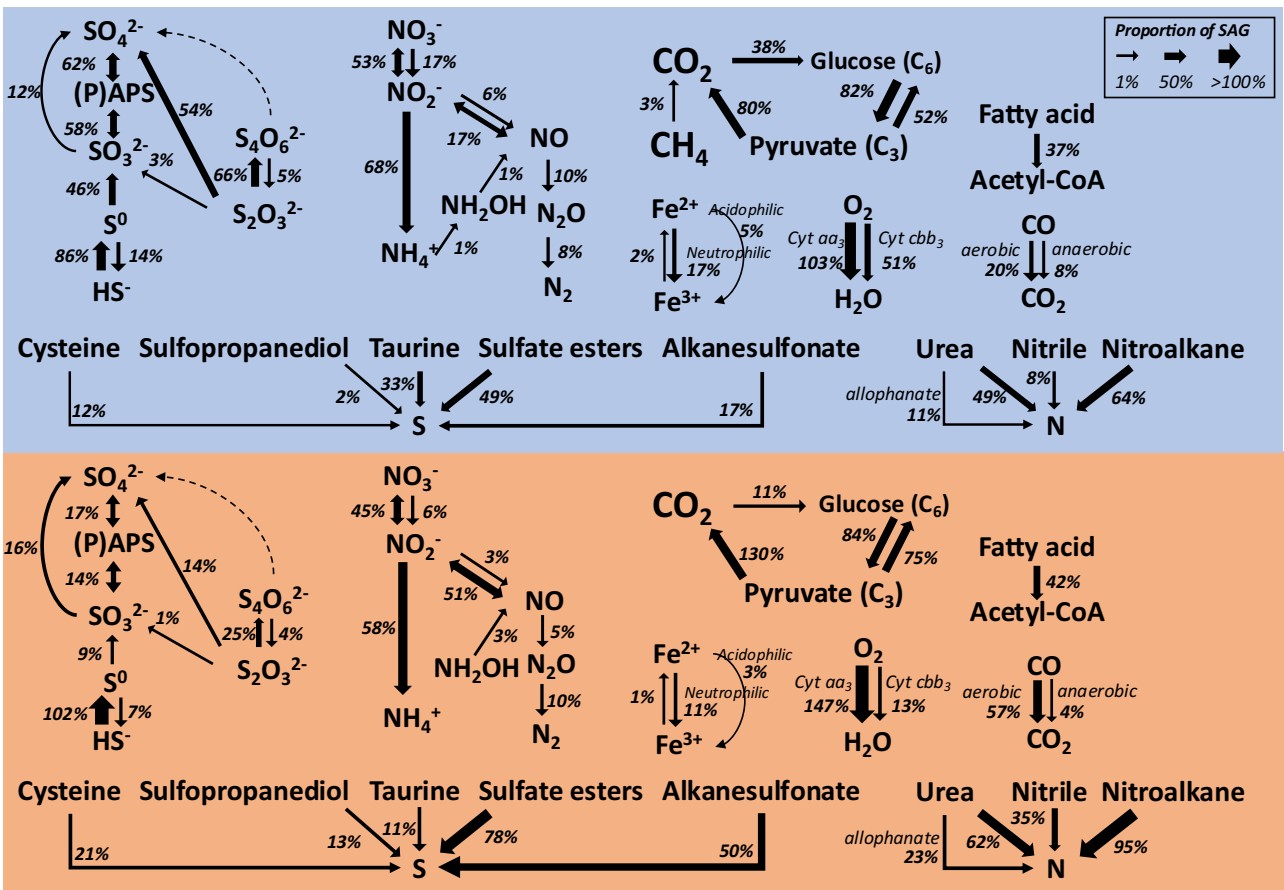

**Fig. 4 | Metabolic potential inferred from SLM SAGs.** The proportion of SLM SAGs involved in a metabolic reaction were calculated to correct for incomplete genome recovery[36] and correlate with the arrow thickness. Note that proportion values exceeding 100% are due to the presence of multi-copy genes in the SAGs. The upper and bottom sections represent metabolic potential of the water column (background in light blue) and sediments (in light brown), respectively. Details on the enzymes involved in the pathways are provided in Supplementary Data 6-7.

been proposed as a means to improve the reliability of functional inference. However, this approach can reduce observed microbial diversity and introduce bias into the community structure. For example, when the analysis was restricted to the 372 SAGs with >50% completeness out of the 1374 SAGs, Actinomycetota (45.6% of the complete dataset) disproportionately increased to 74.5% in the filtered subset. Additionally, such filtering excluded rare but ecologically important taxa, including *Nitrosarchaeum* (completeness <22.8%) and an ammonia-oxidizing Nitrosomonadaceae SAG with 48.2% completeness. Therefore, we included all the 1374 SAGs in the functional analyses, regardless of their genome completeness. We first present a quantitative assessment of metabolic potential for lake_SAGs and sediment_SAGs, followed by a separate analysis of lake_small_SAGs, given that highly incomplete pathways in these smaller genomes may underestimate overall metabolic potential.

None of the SAGs possess genes for photosynthetic pathways, but genes involved in autotrophic $CO_2$ fixation pathways were identified in 38.2% of lake_SAGs and 10.6% of sediment_SAGs (Fig. 4). All the putative autotrophs possessed hallmark genes of the Calvin-Benson-Bassham (CBB) cycle, with the exception of two lake_SAGs belonging to the family UBA6902 (the class Thermodesulfovibrionia), which contained genes for the Wood-Ljungdhal pathway. Though we found no genetic evidence for the hydroxypropionate/4-hydroxybutyrate cycle typical of ammonium-oxidizing archaea[39] in any of the five lake_SAGs of the genus *Nitrosarchaeum*, their CheckM completeness estimates (average of 18.3%) were low. The water column taxa with primary production potential were dominated by the genera *Nitrotoga* (18 SAGs), SURF-13 (7), *Polaromonas* (4), and 39-52-133 (4), whereas in

the sediments, *Nitrotoga* (11 SAGs), SPCO01 (8), UBA4592 (5), and *Polaromonas* (5) were the most abundant autotrophic taxa inferred (Supplementary Data 8).

Multiple copies of genes encoding glycoside hydrolases (GHs) and peptidases were present in nearly all SAGs. There is also evidence for the capacity to conduct Beta-oxidation in 37.3% of lake_SAGs and 41.9% of sediment_SAGs (Figs. 4 and 5). The glycolytic pathway was present in 82.0% of lake_SAGs and 83.5% of sediment_SAGs, with the Embden-Meyerhof (64.8% and 73.0%, respectively), Entner-Doudoroff (ED; 21.3% and 11.8%, respectively), and semi-phosphorylative ED (23.1% and 18.0%, respectively) pathways identified (Supplementary Data 8). The lake_SAGs and sediment_SAGs reveal the prevalence of the TCA cycle (75.5% and 109.9%, respectively), glyoxylate shunt (16.9% and 58.4%, respectively), and gluconeogenesis (51.5% and 75.1%, respectively). Where proportion values exceed 100%, multi-copy genes are present in the SAGs. Genes for fermentation pathways that produce butyrate, lactate, acetate, formate, and ethanol from pyruvate were observed at low frequency in SAGs (Supplementary Fig. 29).

The $aa_3$-type cytochrome c oxidase (*cyt-aa₃*, low affinity for oxygen) was detected in 103.0% of lake_SAGs and 147.2% of sediment_SAGs, whereas the *cyt-cbb₃* type (high affinity for oxygen) appeared in 50.6% of lake_SAGs and 12.7% of sediment_SAGs (Figs. 4 and 5). SAGs of Actinomycetota exclusively possessed *cyt-aa₃*, while the co-occurrence of both cytochrome genes was frequently observed in Pseudomonadota (Supplementary Data 8). Both *Nitrotoga* and SURF-13 possessed only *cyt-cbb₃*.

Neither $N_2$ fixation nor anammox genes were present in the SAG dataset. The complete operon for ammonium monooxygenase

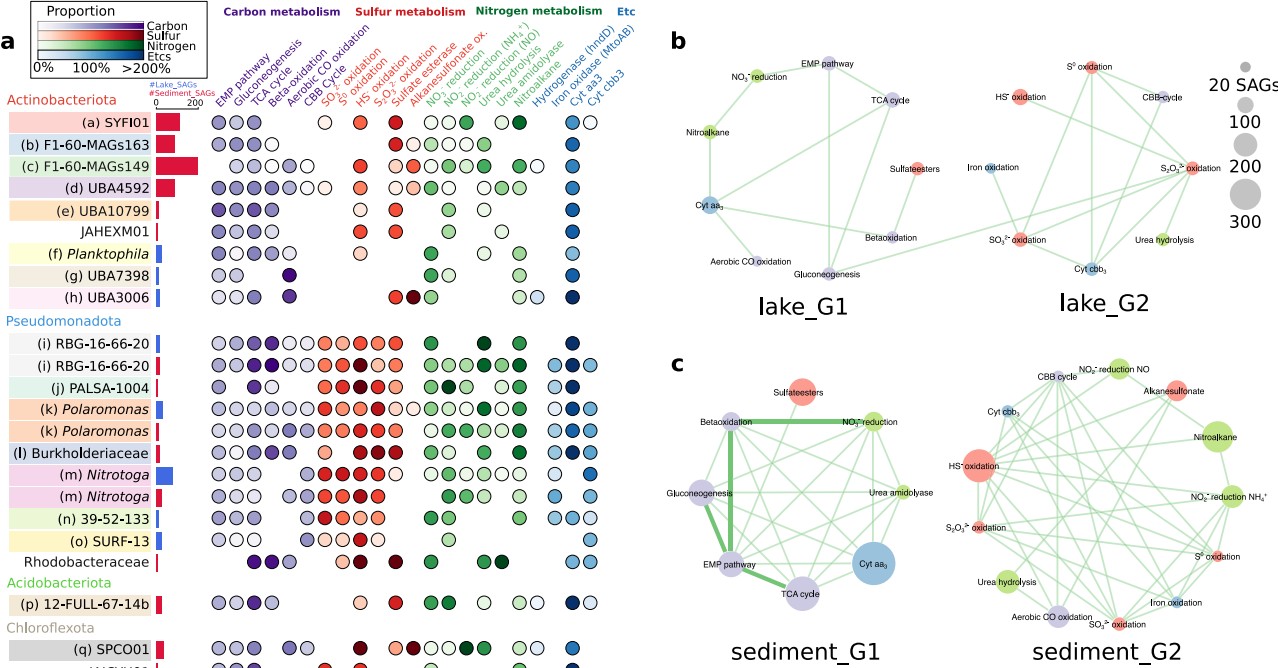

**Fig. 5 | Taxonomic distribution and network representation of metabolic potential. a** The darker the circle shading, the higher the proportion of SAGs responsible for a metabolic potential. The SAGs in a genus were divided into two sets: the water column- (blue horizontal bar for # of SAGs) and sediment (red horizontal bar for # of SAGs). The sets with <9 SAGs were omitted in this figure to simplify presentation. Networks were constructed using co-occurrence of every possible pair of metabolic genes across the SLM SAGs. The upper and bottom networks were derived from SAGs of the water column (**b**) and sediment (**c**). The node size and edge thickness correlate with the proportion of metabolic potential and with the inverse magnitude of the hypergeometric *E*-value for the co-occurrence, respectively. Note that edges are shown only when their *E*-values are less than 0.1. The networks were visualized using CoNet and ClusterViz in Cytoscape. Further details on the chemical reactions of the metabolic pathways are provided in Supplementary Data 6 and 7.

(*amoCAB*) was detected in a lake_SAG classifying within the family Nitrosomonadaceae, but not in the five *Nitrosarchaeum* SAGs (Supplementary Data 8)[39]. Hydroxylamine oxidation genes were observed in 0.9% of lake_SAGs (*Polaromonas*) and 3.1% of sediment_SAGs (mostly the genus 12-FULL-67-14b). The operon for dissimilatory nitrate reductase (*narGHI*) was identified in 53.3% of lake_SAGs and 44.7% of sediment_SAGs. Periplasmic nitrate reductase (*napAB*) was detected in 16.9% of lake_SAGs and 5.6% of sediment_SAGs. Reversible nitrite reductase (16.9% lake_SAGs and 50.6% sediment_SAGs), nitrite reductase (6.2% and 2.8%, respectively), nitric oxide reductase (9.7% and 5.3%, respectively), and nitrous oxide reductase (8.0% and 9.6%, respectively) genes involved in denitrification were also present. Ammonifying nitrite reductase for dissimilatory nitrate reduction to ammonium (DNRA) was common to 67.5% of the lake_SAGs and 58.1% of sediment_SAGs. Nitrogen mineralization pathways were highly represented in the dataset (113.3% of the lake_SAGs and 151.5% of the sediment_SAGs; Supplementary Note 4).

Many of the genes encoding enzymes involved in reverse dissimilatory sulfate reduction (rDSR) were abundant in the SAG dataset (Fig. 4): sulfate adenylyltransferase (62.2% lake_SAGs and 17.4% sediment_SAGs), adenylylsulfate reductase (57.7% and 13.7%, respectively), dissimilatory sulfur oxidase (46.2% and 9.3%, respectively), sulfite oxidase (12.4% and 16.5%, respectively), and sulfide oxidase (86.2% and 101.5%, respectively). Phylogenetic analysis confirmed the *dsrA* genes from SLM SAGs are more closely related to the genetic form of *dsrA* possessed by sulfur-oxidizing lithotrophs, and more distantly related to the version involved with sulfite reduction (Supplementary Fig. 30). No other genes for dissimilatory sulfate reduction were present in the water column and surficial sediment of SLM. The metabolic potential for thiosulfate oxidation was shown by genes for the SOX complex (54.2% lake_SAGs and 13.7% sediment_SAGs), while those for thiosulfate oxidases were less abundant (2.7% and 0.9%, respectively). Genes

encoding enzymes for the oxidation of thiosulfate into tetrathionate (65.7% lake_SAGs and 24.8% sediment_SAGs) are more abundant than those catalyzing the reverse reaction (5.3% and 3.7%). The rDSR and SOX complexes occurred exclusively in Pseudomonadota, whereas sulfide oxidase was widely distributed across diverse phyla (Fig. 5a). Sulfur mineralization pathways were encoded in 95.6% of lake_SAGs and 141.3% of sediment_SAGs (Supplementary Note 4).

A proportion of SAGs contained genes involved in iron cycling and trace gas metabolism (Fig. 4). Three types of iron oxidase were detected: neutrophilic iron oxidase (1.8% lake_SAGs and 1.6% sediment_SAGs), probable neutrophilic iron oxidase (*mtoAB*; 15.1% and 10.6%, respectively), and acidophilic iron oxidase (5.3% and 3.1%, respectively). The SAGs of putative iron oxidizers were mostly Pseudomonadota, but five were affiliated with Acidobacteriota. Genes for iron reductases and probable iron reductases (*mtrABC*) were found in 1.8% of lake_SAGs and 0.6% of sediment_SAGs. Only three lake_SAGs belonging to the genus Methylobacter_A possessed aerobic methane oxidation genes (*pmoCAB*). Genes involved in methanogenic pathways were absent in the SAG dataset, while genes encoding C-P lyase were detected in two lake_SAGs and two sediment_SAGs. The potential to oxidize hydrogen was encoded in 7.1% of lake_SAGs and 2.2% of sediment_SAGs that possessed at least one of the following hydrogen oxidases: *hoxHFUY*, *hndABCD*, *hyaABC*, and *mbhLKJ*. Aerobic CO oxidases (*coxLMS*) were present in 411 SAGs (77.3% lake_SAGs and 96.6% sediment_SAGs). Phylogenetic analysis of CoxL sequences (Supplementary Fig. 31) identified 216 SAGs (19.5% lake_SAGs and 57.1% sediment_SAGs) that encode the form I-like CoxL; a marker of the molybdenum-containing carbon monoxide dehydrogenase (Mo-CODH) used to derive energy from CO oxidation[40]. The form I-like *coxL* was only found in SAGs classifying within the phyla Actinomycetota (159 SAGs), Pseudomonadota (37 SAGs), and Chloroflexota (20 SAGs).

Network constructs using co-occurrence of biogeochemically relevant genes from SAGs produced four metabolic guilds in SLM's water column and sediments with a distinct functional association (Fig. 5b, c). The water column guild lake_G1 has characteristics that include aerobic (Cyt-$aa_3$) and nitrate respiration with catabolism of monosaccharides (EMP and TCA), fatty acids (beta-oxidation), carbon monoxide, and nitroalkane. The lake_G2 guild contains the potential for lithotrophic sulfur oxidation and probable iron oxidation (MtoAB) coupled to Cyt-$cbb_3$, urea hydrolysis, and carbon fixation. The guild sediment_G1 is inferred to be capable of using urea as a N source (urea amidolyase) and resembles lake_G1 devoid of CO oxidation and nitroalkane oxidation. The guild sediment_G2 is similar to lake_G2, but also includes nodes associated with CO oxidation, DNRA, denitrification, nitroalkane oxidation, and alkane sulfonate oxidation. Guilds lake_G2 and sediment_G2 are almost exclusively composed of Pseudomonadota SAGs, whereas lake_G1 and sediment_G1 contain various taxonomic groups.

The lake_small_SAGs (<0.2 μm fraction) exhibited significantly smaller genome size (1,279,069 bp on average) compared to lake_SAGs from cells >0.2 μm (2,837,467 bp) and sediment_SAGs (2,979,839 bp) (Welch's two-tailed t-test, $P < 0.05$). Because the genes and metabolic pathways in these genomes are highly streamlined, combining lake_small_SAGs with lake_SAGs would underestimate the proportion of genomes for metabolic potential. Therefore, the proportion of genomes with a specific metabolic potential for lake_small_SAGs was separately estimated, considering the incompleteness of the SAGs[36]. In contrast to the higher metabolic complexity observed in lake_SAGs and sediment_SAGs, lake_small_SAGs showed a marked deficiency in genes involved in carbon, nitrogen, sulfur and iron cycling (Supplementary Fig. 27 and Supplementary Data 8). Although genes associated with aerobic energy conservation through glycolysis or beta-oxidation were detected, these pathways were predominantly found in non-Patescibacteria SAGs (31 out of 190 lake_small_SAGs). Aside from limited sulfur oxidation potential (7% of lake_small_SAGs for sulfide and 3% for thiosulfate), no complete pathways for the oxidation of inorganic compounds (e.g., nitrification and ferrous oxidation) were detected. Although the nitrite reductase gene (nirK) was identified in both Patescibacteria and non-Patescibacteria, the complete pathway for denitrification was not found.

The phylum Patescibacteria (i.e., Candidate Phyla Radiation), which comprised the majority of lake_small_SAGs (159 out of 190 SAGs) and a smaller subset of lake_SAGs (43 out of 428 SAGs), possessed highly streamlined genomes averaging 1.15 Mbp in estimated size (Supplementary Fig. 32). These genomes lack complete biosynthetic pathways for nucleotides, most amino acids, and lipids (Fig. 6 and Supplementary Data 8). Instead, genes for nutrient uptake and degradation that could complement these metabolic deficiencies were widely distributed in the Patescibacteria SAGs. For example, genes involved in DNA uptake that include type IV pilus assembly proteins (K02654, 125.5%; K02662, 160.1%; K02669, 142.8%) and competence proteins ComEC (K02238, 156.9%) and ComFA (K02242, 111.4%) were overrepresented when the proportion of genomes was estimated considering genome incompleteness (33.8% on average) for the Patescibacteria SAGs. Both exodeoxyribonuclease III (K01142, 97.3%) and TatD DNase (K03424, 149.1%), involved in DNA degradation, were also highly prevalent among the Patescibacteria genomes. The gene encoding peptide/nickel transport system substrate-binding protein (K02035), responsible for oligopeptide uptake, was present at a high genomic proportion (160.1%). A wide array of peptidases were detected, including ClpP (K01358, 120.8%) and DegP (K04771, 86.3%). Genes involved in the oxidation of inorganic electron donors (e.g., sulfur, nitrogen, iron, and hydrogen) appear to be underrepresented in Patescibacteria from SLM. Instead, genes that utilize external carbohydrates as an energy source were frequently observed. These included a multiple sugar transporter system (K02027, 124.0%), glycosidic

hydrolases like alpha-amylase (K07405, 43.9%), glucoamylase (K01178, 31.4%), and beta-glucosidase (K05349 and K05350, 31.4%). Interestingly, phosphofructokinase, a key enzyme in the energy investment phase of glycolysis, was not detected in any Patescibacteria SAGs. On the other hand, pyruvate kinase (K00873), which is involved in the energy payoff stage of glycolysis, was detected in 30 Patescibacteria SAGs. Complete pathways for the TCA cycle and beta-oxidation were not observed. A complete electron transport chain was not detected, although cytochrome $bo_3$ oxidase was found in six SAGs of the genus UBA1550 (the class Paceibacteria) and one SAG of the genus JAICHO01 (the class Saccharimonadia). In contrast, genes encoding subunits of the F-type ATPase (K02108-K02115) were common, with the proportion of genomes ranging from 62.8% to 92.6%. Intriguingly, transaldolase, a key enzyme in the nonoxidative pentose phosphate pathway that serves as a metabolic shunt of glycolysis, was scarcely found in a few Patescibacteria SAGs. Genes encoding lactate dehydrogenases were found in 24 Patescibacteria SAGs (21 with the D-lactate form, K03778; three of the L-lactate form, K00016). Pyruvate fermentation pathways to ethanol or acetate were not complete in any of the individual SAGs, although 38 SAGs possessed genes encoding acylphosphatase (K01512, 35 SAGs) or acetate kinase (K00925, 3 SAGs). Additionally, four SAGs possessed the phosphoketolase gene (K01621), enabling conversion of xylulose-5P to acetyl-phosphate. Genes encoding superoxide dismutase (K04564, K00518, and K04565 for SOD2) that play a critical role in protecting bacteria from free radicals in oxic environments were found in 85 Patescibacteria SAGs at a genomic proportion of 133.4%. Haemolysin (K11068) was detected in a small fraction (3.1%) of the SAGs. Finally, CRISPR spacers that are the genetic footprints of past viral infections were present in 65 SAGs, averaging 1.9 copies per SAG.

## Discussion

SLM is below sea level, is proximal to the oceanographic system (~50 km from the current grounding line), and has sediments containing evidence for recent marine interactions[12], yet the relatively low chloride concentration in sediment pore waters indicates deposition by an upstream freshwater catchment[41]. This conclusion aligns with our high-resolution phylogenomic analysis of single-cell populations in SLM's water column and sediments, which reveals that nearly all genomes are most closely related to those documented in terrestrial aquatic ecosystems. Rarely did they form sister groups with genomes recovered from marine environments (Supplementary Figs. 6–25), providing low support that its extant species evolved from marine taxa.

The SLM SAGs form cohesive phylogenetic clusters to the exclusion of any of the 596,859 genomes available in release 220 of GTDB (Fig. 3a and Supplementary Figs. 7–25) and encode ORFs with low similarities to known proteins in the NCBI non-redundant database (Fig. 3c). The phylogenetic and genetic differences are mirrored at the functional level across the most abundant genera (Supplementary Figs. 7–25 and Supplementary Data 10), which is likely an evolutionary outcome of microbial isolation in Antarctic subglacial environments. Average ANI values of 77.0% (Fig. 3b) are below the cutoff of 95% commonly used to demarcate prokaryotic species[42], providing a glimpse of the unknown biodiversity under the Antarctic ice sheet and implying that the 1372 SAGs likely represent new species candidates or higher novel taxonomic groups. A further consideration for populations in SLM is the low energy conditions and their slow growth rates[20], which should fundamentally influence their evolutionary rates[43]. As such, the level of diversification observed in SLM SAGs would not be expected if derived from populations introduced to this region ~6300 years before present[12]. Data based on 16S rRNA gene sequences[17] showed that approximately half of the taxa detected in SLW were also present in SLM, suggesting a common upglacier source of microbial innocula to the lakes and implicating co-transport with subglacial

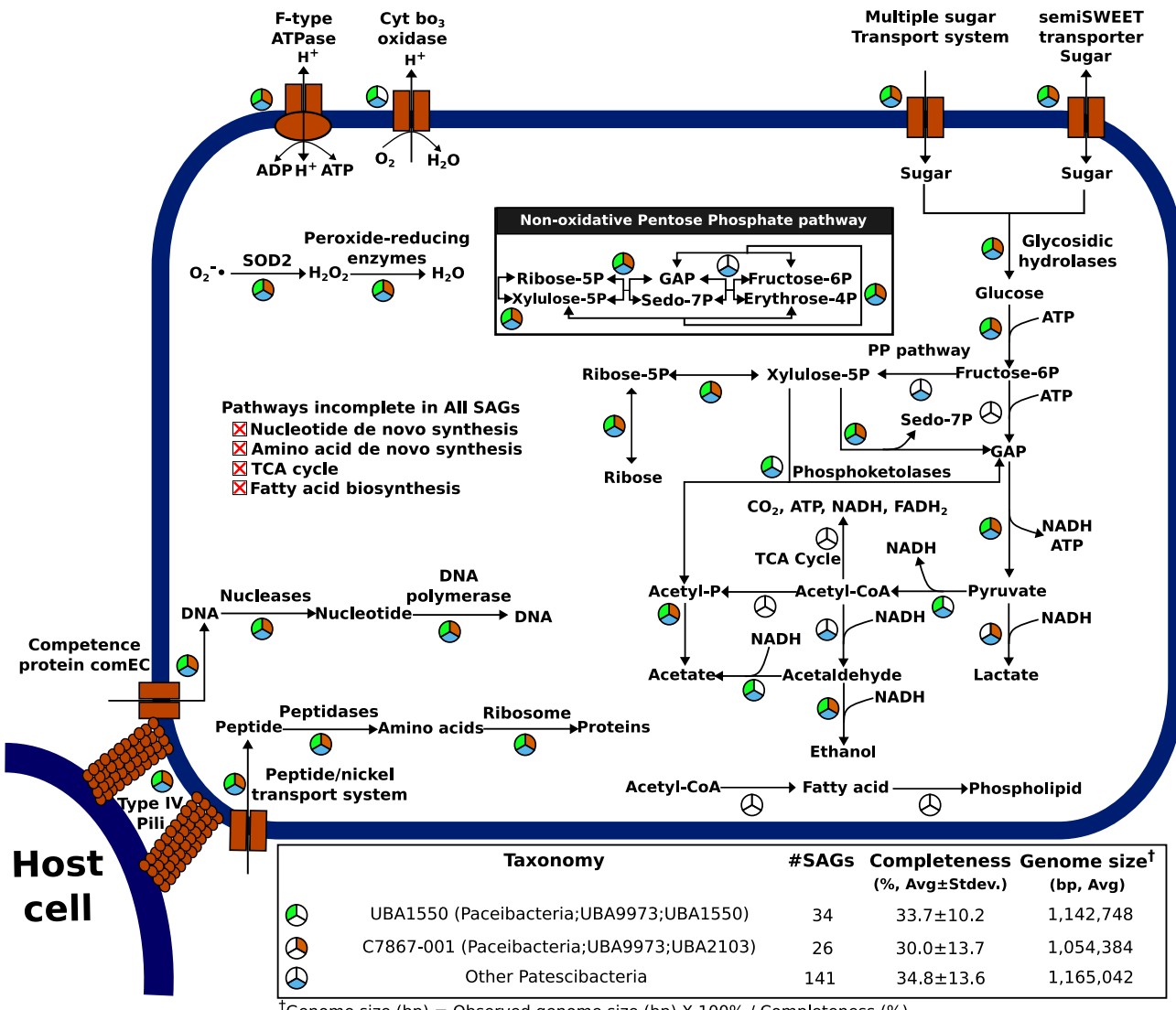

**Fig. 6 | Schematic representation of metabolic potential for Patesci-bacteria in SLM.** The presence or absence of metabolic pathways was determined based on KEGG annotations and the criteria outlined in Supplementary Data 6 and 7. Green-filled sectors represent pathways detected in one or more SAGs of the genus UBA1550, which is the most abundant genus of Patescibacteria in SLM. Red-filled sectors indicate pathways present in one or more SAGs of the genus C7867-001, the second most abundant genus. Blue-filled sectors denote pathways found in one or more genomes of the remaining Patescibacterial SAGs. Unfilled circles indicate pathways that were not detected in any patescibacterial SAGs. An inset provides a detailed view of the non-oxidative pentose phosphate pathway, spanning from fructose-6-phosphate to xylulose-5-phosphate. Sedo-7P stands for sedoheptulose 7-phosphate. Source data are provided as a Source Data file.

water and sediment as a dispersal mechanism. If SLM is a component of a metacommunity that is linked via subglacial hydrological systems, then the upper timeframe for population isolation may correspond to the most recent destabilization of WAIS in the last few million years[44,45]. Evolutionary dating was not performed in this study, as the differences in evolutionary rates between subglacial microorganisms and their nearest neighbors will need to be carefully considered before accurate divergence time estimates are possible (Supplementary Note 5)[46].

Due to the lack of sunlight and photosynthesis in Antarctic subglacial environments, the sole source of oxygen is from ice-entrapped atmospheric gases introduced to subglacial water during basal melting[21]. Our study revealed that both Cyt-*aa3* and Cyt-*cbb3* type cytochrome c oxidases were intimately associated with metabolic guilds in the water column and surficial sediments (Fig. 5), indicating the importance of oxygen as an electron acceptor for facilitating redox reactions in SLM. Specifically, the prevalence of *cyt-aa3* (103.0% of lake_SAGs and 147.2% of sediment_SAGs) over *cyt-cbb3* (50.6% lake_-SAGs and 12.7% sediment_SAGs) is a characteristic consistent with high oxygen availability. Interestingly, SAGs of Actinomycetota only possess the *cyt-aa3* type, while those of Pseudomonadota often have both types, suggesting that the proteobacterial species may have larger tolerances to oxygen availability.

The four metabolic guilds inferred by the co-occurrence network analysis also highlight the role of oxygen as a parameter shaping the microbial community and influencing SLM biogeochemical processes (Fig. 5b, c). Pathways for oxidizing simple sugars (glycolysis pathway in 83.5% sediment_SAGs) and fatty acids (beta-oxidation in 41.9% sediment_SAGs) may be coupled with either aerobic or anaerobic respiration in surficial sediments (sediment_G1 in Fig. 5c). The widespread incidence of genes encoding GHs and peptidases in the SAGs indicates a high potential for the utilization of complex carbon sources including carbohydrates, peptides, organic sulfur (e.g., sulfate esters), and organic nitrogen (e.g., nitroalkane) (Figs. 4 and 5). Both sediment guilds possess the genetic capacity for aerobic respiration but have different preferences in their electron donors (inorganic vs. organic; Fig. 5c). In sediment_G2, ~10% of sediment_SAGs contain genes for

lithotrophic energy generation and fixing carbon dioxide, including the oxidation of a variety of reduced inorganic compounds such as sulfide, thiosulfate, carbon monoxide, and $Fe^{2+}$ using oxygen or nitrate. DOC that is supplied by upward diffusion from sediments[8] and chemosynthesis in the water column (lake_G2 in Fig. 5b) would support organotrophic lifestyles (lake_G1 in Fig. 5b). The presence of genes enabling metabolism of carbon monoxide (sediment_G2, 57.1% sediment_SAGs; lake_G1, 19.5% lake_SAGs) suggests CO oxidation could supplement cellular energy production in the water column and surficial sediment communities.

Although SLM's water column was supersaturated with oxygen at the time of sampling[24], our single-cell genomes exhibited characteristics consistent with flexibility to metabolize under suboxic and anoxic conditions. For example, ~13% of sediment_SAGs contained the gene complement for $Fe^{2+}$ oxidation coupled to Cyt-$cbb_3$ for energy generation (sediment_G2 in Fig. 5c). In the relatively lower oxygen concentration in the surficial sediments, the presence of this pathway may enhance metabolic competition for oxygen over chemical oxidation[47]. However, since a similar proportion of the genomes from iron oxidizers (19.5% of lake_SAGs) also occurred in lake_G2 with Cyt-$cbb_3$, it is unclear if the presence of distinct cytochrome c oxidases has phenotypic consequences that enhance iron-oxidation over a range of oxygen concentrations. Denitrification and fermentation, both of which are known to occur under anoxic conditions, were also detected in both the water column and sediments (Fig. 4 and Supplementary Fig. 29). Their presence may be explained by the formation of anoxic microenvironments[48] within oxygenated habitats. Alternatively, these metabolisms may occur when SLM becomes anoxic due to the depletion of dissolved oxygen driven by chemical oxidation and microbial respiration[49]. While methanotrophic functional genes were abundant and methane oxidation rates were measured in SLW[17,19], our single-cell data from SLM contain only five lake_SAGs affiliated with the genus Methylobacter_A and with the functional complement of genes to oxidize methane. However, it remains possible that rare, yet active taxa contribute to those pathways[50].

None of the SAGs encode genes involved in nitrogen fixation, while genes encoding the complete denitrification pathway are relatively abundant (Fig. 4). This suggests that microbe-mediated nitrogen cycling in SLM may diminish the biogenic N pool[50]. The main sources of nitrogen (e.g., ammonium and nitrate) are presumably from basal ice melt, the underlying sediments, and bedrock comminution[22,51]. Ammonium is believed to be an important electron donor in many subglacial environments[15,16,52], but surprisingly, only one SAG was identified that contains genes for ammonium oxidation (the family Nitrosomonadaceae) (Supplementary Data 1). The putative nitrite oxidase gene (nxrA) was identified via KEGG annotation in only three out of 108 Nitrotoga SAGs, despite this genus being known for nitrite oxidation[53]. Since NxrA shares the same KEGG ortholog K00370 with nitrate reductase (NarG), we compared the three sequences against curated sequences of NarG and NxrA from the nitrogen cycling database NcycDB[54]. The results indicated higher similarity to NarG than to NxrA. BLASTp comparisons further revealed that the sequences are more similar to NarG of the nitrate reducer Sulfuriferula multivorans (88% sequence similarity with WP_124703789.1 in GenBank) than to NxrA of the experimentally verified nitrifying Candidatus Nitrotoga arctica (23% similarity with WP_239795842.1). Hence, the metabolic capacity for nitrite oxidation is not exhibited by the 108 Nitrotoga SAGs we characterized from SLM. Instead, their gene complement suggests they conserve energy organotrophically or use a variety of inorganic electron donors, including reduced sulfur compounds, $Fe^{2+}$, and CO, to grow lithotrophically (Fig. 5a). To evaluate the overall metabolic potential for nitrite oxidation in SLM, we screened all the proteins of the 1,374 SAGs against NxrA sequences from NCycDB. This analysis identified one nxrA gene in the single-cell SLM_LV1_0208-B14, which is affiliated with the family Opitutaceae in the phylum Verrucomicrobiota

(Supplementary Data 1). A subsequent BLASTp comparison of this sequence revealed 70.8% similarity to the nxrA gene of the known nitrite oxidizer Thiocapsa KS1[55] (CRI68048.1 in GenBank), and only 24.2% similarity to the narG gene of the nitrate-reducer Opitutus terrae[56] (WP_012374288.1 in GenBank), a member of Opitutaceae. These results suggest the possible presence of nitrite oxidizers in SLM.

Chemosynthetic primary production could be the most important process supporting heterotrophic life in dark subglacial environments[16]. Among the 1,374 SAGs, 77 were identified that possess the Calvin-Benson-Bassham (CBB) cycle. These putative autotrophs were found in 11% of sediment_SAGs and 38% of lake_SAGs (Fig. 4), suggesting that both habitats have the potential to support primary production. Although some primary producers were exclusive to each habitat (e.g., the genera SURF-13 and 39-52-133 in the water column, and SPCO01 and UBA4592 in the sediments), Nitrotoga and Polaromonas were common to the water column and sediments (Fig. 5a). In addition, we observed a strong association between the CBB cycle and the $cbb_3$ type cytochrome c oxidase (Fig. 5b, c). Cells with the cyt-$cbb_3$ type (27 out of 102 SAGs; 26.5%) were more likely to possess the CBB cycle than those with the cyt-$aa_3$ type (26 of 610; 4.3%). For example, Nitrotoga, the most abundant autotrophic genus in the SAG datasets (33 out of 77 cells), was exclusively associated with the cyt-$cbb_3$ type (Supplementary Data 8). SAGs containing both the cyt-$cbb_3$ type and the CBB cycle were frequently linked to the oxidation of reduced sulfur compounds (Fig. 5b, c). These findings suggest that chemolithoautotrophs harboring the cyt-$cbb_3$ type may become more competitive under low-oxygen conditions, where the activity of the $cbb_3$-type cytochrome c oxidase is optimal. Only 7 out of the 77 SAGs (6 are Nitrotoga) possessed both the CBB cycle and nitrate reductase, indicating that carbon fixation can occur when the SLM environment becomes anoxic. The major primary producers identified in this study (e.g., 31 Nitrotoga SAGs, 9 Polaromonas SAGs, and 7 SURF-13 SAGs) are all considered mixotrophs since they possessed both organotrophic pathways (e.g., EMP pathway, TCA cycle, or β-oxidation) and the CBB cycle (Fig. 5a). Interestingly, mixotrophic Polaromonas species are also considered major contributors to primary production in groundwater environments[57], but not all are able to fix carbon dioxide (e.g., supraglacial Polaromonas species relying on existing organic carbon for growth while using carbon monoxide as an electron donor[58]). These taxa are predicted to be capable of oxidizing reduced sulfur compounds and $Fe^{2+}$ through both aerobic and nitrate-based anaerobic respiration, representing a range of trophic strategies spanning from heterotrophy to chemoautotrophy. This metabolic versatility could provide microbes in SLM with the capacity to physiologically adapt to the dynamic changes of oligotrophic subglacial lake environments associated with periodic fill-drainage events.

Approximately 15% of the 1,374 SAGs discovered in SLM belonged to Patescibacteria, and the majority of these cells were isolated from 0.2 μm pore size filtrates that indicate their ultrasmall cell size. Patescibacteria also exhibited the smallest genome sizes compared to other bacterial phyla present in SLM (Supplementary Fig. 32). Both the abundance and genome reduction indicate that Patescibacteria are competitive to survive in the oligotrophic SLM environment. Genome reduction has evolutionarily resulted from streamlining genomes involved in energy-demanding molecular functions under natural selection[59]. Therefore, small microbes like Patescibacteria suffer from leaky metabolisms in terms of energy conservation and biosynthesis. Comparative genomic analysis in this study revealed that Patescibacteria in SLM possess rudimentary energy metabolisms (Fig. 6), which is consistent to other studies[60–65]. Since the complete pathways for the TCA cycle and the electron transport chain were not detected, Patescibacteria in SLM probably ferment organic substrates to conserve energy. These bacteria are likely to uptake sugar molecules via transporters and degrade them using glycoside hydrolases. However, metabolizing glucose to pyruvate seems not possible since

phosphofructokinase was not present in the Patescibacteria SAGs. While the possibility that salvaging glucose-6-phosphate into fructose-6-phosphate and glyceraldehyde-3-phosphate can be proposed[60], this metabolic shunt would be scarce since the non-oxidative pentose phosphate pathway was complete in only a few Patescibacteria SAGs. Given the presence of pyruvate kinase in 30 Patescibacteria SAGs, ATP generation through the second half of glycolysis could occur if glyceraldehyde-3-phosphate or other three-carbon glycolytic intermediates are supplied from unknown pathways or external sources[63,66]. In addition, the pathway that ferments pyruvate to acetate is unlikely to occur since the acetate kinase gene was only found in three Patescibacteria SAGs. Although various hypotheses have been suggested (e.g., capturing protons from host cells and generating ATP using F-type ATPase)[63], the cultivation of Patescibacterial strains is necessary to elucidate their energy metabolisms[67]. Given their limited biosynthetic capabilities (e.g., amino acids, lipids, and nucleotides), Patescibacteria likely rely on close associations with co-occurring microbes in SLM to sustain their survival. The widespread presence of genes encoding type IV pilus assembly proteins supports their epi-symbiotic potential for cell-to-cell interactions with host bacteria. Additionally, the high abundance of competence proteins, nucleases, and peptidases may compensate for their metabolic deficiencies by enabling the breakdown of macromolecules and the acquisition of essential building blocks for growth. These metabolic dependencies highlight the communal lifestyle of the subglacial Patescibacteria. However, the rare detection of haemolysin genes suggests that a small proportion of these bacteria could adopt a parasitic strategy[65]. In contrast to Patescibacteria isolated from low-oxygen groundwater samples[61], the frequent occurrence of CRISPR spacers in patescibacterial SAGs implies their potential mutualistic role, in which they may act as decoys to divert phage infections away from their host cells[63]. Collectively, Antarctic subglacial Patescibacteria likely exhibit a continuum of symbiotic relationships, ranging from communal to parasitic and mutualistic interactions, as previously reported in freshwater Candidate Phyla Radiation[64].

In conclusion, the metabolic pathways we have inferred from SLM genomes reveal that microorganisms in the subglacial ecosystem have the capacity to conserve energy through the oxidation of various organic and inorganic compounds (e.g., reduced sulfur and $Fe^{2+}$) with oxygen, nitrate, or $Fe^{3+}$ as electron acceptors. Oxygen availability has played a key role in shaping the microbial community and influences the metabolic processes occurring in SLM. Comparative analysis of functional and evolutionary divergence in the single-cell genomes from SLM supports a contention for genetic isolation of the subglacial populations from those in contemporary surface and marine biomes. Despite a marine incursion at the current location of SLM 6300 years ago[12], its SAGs are not related to those of contemporary marine microorganisms. If the microorganisms in the lake have an upglacial source from the ice sheet interior, then SLM may be a constituent of a metacommunity that has evolved in relative isolation beneath WAIS. Examining organismal and genetic flow between SLM, SLW, and the other subglacial lakes in this region could determine if there is metacommunity structure across discrete basins beneath the Antarctic ice sheet.

## Methods

### Sampling of lake water and sediment from SLM
The lake was directly accessed through a ~0.4 m diameter borehole melted through 1,087 m of ice with an environmentally clean, hot water drilling system[16,23]. Using a Large Volume Water Transfer System (WTS-LV; McLane Research Laboratories Inc.), lake water was sampled from the first cast (LV1) at mid-depth of the ~15 m water column and concentrated in-situ using filters with pore sizes of 3.0, 0.8, and 0.2 μm[23]. A sediment catcher attached to the base of the WTS-LV frame was used to collect bulk surface (BS) sediment. Sediment cores were obtained using a multicorer (UWITEC)[68]. The second core of the first

multicore cast (MC1B) was horizontally sectioned at 2 cm intervals, and inner portions from each interval were collected for molecular and single-cell genome analysis. All samples were collected aseptically and immediately cryopreserved at −80 °C as described previously[16,17].

### Microbial sample treatments and single-cell genomics
Triplicate lake water samples (1 ml each) were supplemented with 5% v/v glycerol, 1× TE buffer (final concentrations) and stored at −80 °C until analysis. Approximately 5 g of sediment sample was mixed with 20 ml sterile-filtered PBS, vortexed for 30 s, and centrifuged for 30 s at 2500 g to remove large particles. The resulting supernatant was treated with the GlyTE buffer as described above. Samples for cell sorting were pre-screened through a 40 μm mesh size cell strainer (Becton Dickinson) and stained with SYTO 9 (5 μM, Thermo Fisher Scientific) for 1 h. Single-cell sorting using fluorescence-activated cell sorting (FACS) was followed by whole genome amplification (WGA) and was carried out at the Single Cell Genomic Center at Bigelow Laboratory for Ocean Sciences (SCGC) as previously described[27]. Briefly, stained and filtered samples were processed on a BD Influx flow cytometer equipped with a 488 nm laser. Sorting gates were configured to isolate single cells while minimizing background signals (Supplementary Fig. 33). The sorted cells were then subjected to single-cell whole genome amplification for downstream genomic analysis.

The libraries were prepared using the TruSeq Nano DNA High Throughput Library Prep Kit (Illumina) and following the manufacturer's protocols. The products generated were purified and PCR-amplified. The library pool was quantified using a KAPA qPCR library quantification kit (KAPA Biosystems, Wilmington, MA, USA), and library quality was assessed using the Agilent Technologies 4200 TapeStation D1000 ScreenTape (Agilent technologies). Sequencing was carried out using the HiSeq X ten system (Illumina).

### Genome assembly and annotation
The Illumina sequencing raw reads were trimmed and filtered using Trimmomatic v.0.39 with the following parameters: ILLUMINACLIP:-TruSeq3-PE.fa:2:30:10:2:keepBothReads LEADING:0 TRAILING:5 SLI-DINGWINDOW:4:15 MINLEN:36. Low complexity reads containing less than 5% of any nucleotide were discarded[27]. The remaining reads were mapped to a human reference genome (GRCh38) using BWA v0.7.17[69], and reads mapping with 95% or more similarity were considered human DNA and removed. To improve assembly quality[70], the reads were normalized in silico using kmernorm v1.05 (https://sourceforge.net/projects/kmernorm/) with the following options: -k 21 -t 30 -c 3.

Genome assembly was conducted using SPAdes v3.13.0 and the following parameters: --sc --careful[70]. The 100 bp end of each resulting contig was trimmed, and contigs shorter than 2 kbp were removed. Both completeness and contamination of assemblies were evaluated by CheckM v1.1.3[28], and their taxonomic positions were determined by GTDB-Tk v2.4.3 with GTDB r220[29,71]. After discarding genomes with either >5% contamination or failure of taxonomic assignment, the remaining genomes were further screened using the NCBI's Foreign Contamination Screen tool suite to remove contaminant sequences[72]. Then, the quality and taxonomy of the genomes were reexamined by running CheckM and GTDB-Tk. Structural and functional annotations were conducted using Prokka v1.13[73]. To predict gene function, translated protein sequences were BLASTP-searched against KEGG DB release 2020-03-23[74], and KEGG orthologs (KO) of top-scored subject sequences were assigned to query proteins when alignments between query and subject sequences had >30% sequence similarity and >70% query coverage[75]. To distinguish NxrA from NarG that share the same KEGG ortholog (K00370), protein sequences from the SAGs were searched using DIAMOND v2.1.11.165 against curated NxrA and NarG sequences in NCycDB[54]. To detect genes involved in iron metabolism and viral sequences, the single-cell genome sequences were analyzed using FeGenie v1.0[76] and VirSorter v1.0.6[33], respectively. For

sulfur-metabolizing genes absent in the KEGG database, BLAST searches were conducted using reference sequences from the literature[77]. The average nucleotide identity of genomes compared was calculated using orthoANIu v1.2 and default options[78].

## Phylogenomic reconstruction of single-cell genomes

To determine the phylogeny of the bacterial SAGs, 120 marker genes were aligned with the GTDB multiple sequence alignment using GTDB-Tk v2.4.3[29], and a phylogenomic tree was reconstructed using RAxML v8.2.12[79], 100 nonparametric bootstrap replicates, and a PROTGAMMAWAG substitution model[71]. The bacterial tree was rooted by the most recent common ancestor of the Candidate Phyla Radiation (i.e., Patescibacteria), which are deep-branching phyla in domain Bacteria[80]. An unrooted phylogenomic tree of the archaeal SAGs was similarly constructed but using an archaeal GTDB alignment consisting of 53 marker genes. The phylogenetic trees generated were visualized using either Dendroscope v3.5.10[81] or ETE v.3.1.3[82].

## Estimating taxonomic and functional diversity

CHAO1 was used to estimate taxonomic richness and functional richness for the genomic occurrence of every KEGG ortholog. Since the ANI calculation between the incomplete SAG genomes often failed, we used GTDB genera or higher rank taxonomic annotation as operational taxonomic units instead of de novo SAG clusters determined by the ANI. For both taxonomic and functional richness, the calculations to generate rarefaction curves with 95% confidence intervals using 100 iterative randomizations were made by in-house Python scripts.

## Comparing 16S rRNA gene sequences from SAGs to amplicon data from SLM, SLW and WGZ

Sequences of the V4 region for 16S rRNA genes from SLM (bioproject PRJNA790995), SLW (PRJNA244335), and WGZ (PRJNA869494) were retrieved from NCBI[17]. The sequences were trimmed using Cutadapt v4.2[83] and analyzed with DADA2 v1.25.2. This included quality filtering, error correction, chimera removal, and taxonomic annotation with SILVA DB v138.1[84]. The resulting ASVs were BLASTn-searched against the 16S rRNA gene sequences from SAGs with thresholds of >70% query coverage, >200 bit-score, and <1% sequence dissimilarity. A Venn diagram was used to compare the number of ASVs shared among the three sites. The ASVs were further clustered into OTUs with >97% sequence similarity using VSEARCH v2.24[85], compared to the 16S rRNA gene sequences of SAGs by Bray-Curtis dissimilarity, and visualized with nonmetric multidimensional scaling (NMDS) of samples using vegan v2.6-4 in R v4.3.1. Differences in community structures between microbial habitats, as well as between 16S rRNA and SAGs, were statistically assessed using the PERMANOVA method with 10,000 iterations.

## Evaluation in the divergence of SAGs and functional genes

For all possible pairs between SAGs and the GTDB representative genomes, ANI values were calculated using orthoANIu[78]. When less than 10% of the SAG sequence aligned with any GTDB genomes, the SAG was considered a 'No match'. Phylogenetic distance of predicted SAG proteins was estimated by clustering sequences with a 95% similarity cutoff using CD-HIT v4.8.1[86]. The longest translated amino acid sequence from each protein cluster was BLASTp-searched against the NCBI NR DB released on 27 September 2021. The sequences with no significant hits at the *E-value* of 10 were excluded in the calculation of the mean and mode. To compare SAG protein sequence divergence from known homologous proteins, the longest amino acid sequence from each cluster and the best hit resulting from each BLASTp search were analyzed. To compare sequence divergence in the SAGs from SLM, SAG data derived from previous studies were analyzed using the same procedure. The following criteria were used to select appropriate data: (1) > 30 SAGs were produced to allow robust statistical analysis; (2) SAGs publicly available at the time of data survey in November,

2021; (3) single cells were randomly chosen and sequenced; (4) the study site was a natural habitat; and (5) in order to avoid self-hits, environmental studies whose protein sequences were already recorded in the NCBI NR DB were not chosen. Screening of metadata and sequence comparisons (~100% identity) showed that the only dataset fulfilling the above criteria was a single-cell genomic study of seawater beneath Ross Ice Shelf[35] which was chosen for the analysis. Genome quality was evaluated using CheckM with the threshold of <5% contamination. ANI values between RIS and GTDB genomes were also calculated using orthoANIu. Since the RIS's SAGs were included in the GTDB r220, self-comparisons were excluded during ANI calculations. The proteins of RIS's SAGs were structurally annotated, clustered, and compared with the extant proteins as described above.

## Phylogenomic tree reconstruction for the 19 most abundant genera

For genera with a relative abundance >1% of the SAGs, the GenBank assembly accessions of genomes were extracted from the GTDB metadata and joined with those retrieved from NCBI. Given the possibility that a SAG may represent a novel genus, we conducted a broader taxonomic sampling that included all GTDB genomes from the same family as each SAGs and an outgroup phylogenetically external to the genomes analyzed. A rooted phylogenetic tree was constructed using FastTree v2.1.11[32] of *de_novo_wf* of GTDB-Tk, and the node confidence was evaluated using the Shimodaira-Hasegawa test[87]. Based on the node of the tree representing the most recent common ancestor of the genus of interest and its neighboring genera, a subtree was extracted using ETE. In the subtree, the monophyly of the SAGs was evaluated using the genealogical sorting index (GSI) with 10,000 permutations[34]. Protein sequences from the GTDB genomes identified as sister taxa in each phylogenomic tree were functionally annotated using BLASTp searches against the KEGG database, as described above.

## Extracting environmental sources of public genomes from metadata

The environmental source of genomes deposited in public databases such as NCBI is often omitted and described using non-standardized terminology. To extract the source environments from metadata for large set of genomes, we employed a two-steps procedure. At first, habitat-describing keywords (Supplementary Data 3) were searched against genomes in all bioprojects and biosamples of NCBI. When a keyword is matched, a corresponding habitat "flag" turned on (e.g., the flag for *freshwater* includes the keywords *lake* and *lentic*). The combination of the flags with 'on' was used to classify habitats for each genome for which metadata were available (Supplementary Data 4). Referring to GOLD Ecosystem Classification paths[88] and a previous work[62], we designed habitats, habitat flags, and their associated keywords.

## Sequence-based determination of metabolic function

We identified the phylogenetically distinct forms of dissimilatory sulfite oxidase (large subunit, DsrA) that catalyze sulfide oxidation or sulfite reduction, allowing prediction of the likely reaction direction catalyzed by the queried gene products. DsrA sequences from cultivated sulfur oxidizers and reducers available in public databases[89,90] were combined with those identified in the SAGs, aligned using ClustalW v2.1[91], and a maximum-likelihood tree was constructed using RAxML with 100 bootstrap replicates and a protein substitution model determined using ModelTest-NG[92]. Based on the tree topology, the DsrA sequences of SAGs were classified as either preferentially catalyzing sulfide oxidation or sulfite reduction. In a similar manner, the form I of aerobic carbon monoxide oxidase (CoxL) that likely functions in CO oxidation was identified by a clear phylogenetic separation from the functionally unknown form II of CoxL. The form I and II sequences were analyzed as described previously[40].

## Determination of metabolic potential encoded in the SAGs

The potential for a SAG to encode a single-step pathway was determined by the presence or absence of genes responsible for that reaction (Supplementary Data 6). For example, a SAG is considered to exhibit sulfite oxidation when one or more of the following genes were found: *dsrAB*, *fsrNC*, and *hdr*-like. A set of genes (e.g., *dsrAB*) was considered present in a SAG when half or more of the genes were detected. Accessory subunits (e.g., *dsrL*) were excluded in these determinations. The potential for a SAG to encode a multiple-step pathway (e.g., glycolysis) was determined by the presence or absence of genes encoding key enzymes. Since the SAG data do not represent complete genomes, an additional condition was that a SAG possess genes for >50% pathway completeness (Supplementary Data 7). Extensive surveys of literature[74,77,93,94] and KEGG metabolic pathways guided efforts that evaluated the genes encoding enzymes involved in key metabolic reactions and multiple-step pathways (Supplementary Data 6-7). The proportion of SAGs with a metabolic pathway of interest was estimated to correct for genome incompleteness using the method described by Acker et al. (2022)[36]. In essence, the observed proportion was adjusted by the ratio between the expected[28] and observed genome sizes.

## Artificial reduction of GTDB genomes closest to SAGs

To simulate the quantitative impact of low genome completeness on metabolic potential, each of GTDB genomes closest to the SAGs based on ANI distance was artificially reduced to match the genome completeness of its corresponding SAG using the following procedure: (1) contigs were randomly shuffled; (2) to preserve operon structure, contigs were selected sequentially from the first until the cumulative genome completeness matched that of the SAG counterpart; and (3) in the final contig, a DNA fragment containing the required number of genes to achieve the genome completeness was randomly selected to pursue equal probability of gene inclusion regardless their positions on the contig. Metabolic pathways of the GTDB genomes were predicted by BLASTP-searching translated protein sequences against KEGG DB release 2020-03-23[74]. KEGG orthologs (KO) were assigned to query proteins when alignments between query and subject sequences had >30% sequence similarity and >70% query coverage[75]. Only 70 out of the 80 metabolic pathways surveyed in this study were analyzed since the remaining 10 were either based on non-KEGG annotations or were entirely undetected in the GTDB genomes. Statistical significance was assessed using the Monte Carlo permutation test (1,000 permutations), with *p*-values corrected for false discovery rate (FDR).

## Genomic co-occurrence network of biogeochemically-relevant genes

For each of the 1,374 SAGs, a profile was prepared that described the presence or absence of key metabolic functions of ecological importance. For each functional category, co-occurrence was examined across the profiles by evaluating the statistical significance of the hypergeometric distribution and a threshold *E*-value of 0.1 using CoNet v1.1.1[95] of Cytoscape v3.9.0. Results for significant co-occurrences were used to construct a network that was divided into individual metabolic guilds using the fast agglomerate algorithm FAG-EC of ClusterViz v1.0.3[96]. The co-occurrence analysis was conducted separately for lake water and sediment SAGs.

## Reporting summary

Further information on research design is available in the Nature Portfolio Reporting Summary linked to this article.

## Data availability

The genome sequence data generated in this study have been deposited in NCBI under the project PRJNA1084198. Source data are provided within this paper. Source data are provided with this paper.

## Code availability

The custom Python scripts used in this study are available at the following repositories: 1) habitat classification of microorganisms based on GTDB and NCBI metadata at https://github.com/kyuinHwang/genome-habitat-classification-pipeline, 2) KO-based inference of metabolic potential in microbial genomes at https://github.com/kyuinHwang/ko2pathway, and 3) microbial genome reduction to specified completeness levels at https://github.com/kyuinHwang/genome-completeness-reduction-pipeline.

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

## Acknowledgements

We thank Dr. Kitae Kim at Korea Polar Research Institute for sharing his knowledge about chemical oxidation and Ms. Soyeon Kim for technical assistance. This work was supported by Korea Polar Research Institute (grant number PE18340 to OSK, PE20130 to OSK, PE25130 to OSK) and the US National Science Foundation, Section for Antarctic Sciences, Antarctic Integrated System Science program as part of the interdisciplinary (Subglacial Antarctic Lakes Scientific Access (SALSA): Integrated study of carbon cycling in hydrologically-active subglacial environments) project (NSF-OPP 1543537 to JCP and NSF-OPP 1543396 to BCC).

## Author contributions

K.M.K.: supervised research, analyzed data, wrote the original draft, and revised the manuscript. K.H.: analyzed data, wrote the original draft, and revised the manuscript. H.L.: revised the manuscript. A.C.: conducted sample analysis processing. C.L.D.: collected samples and revised the manuscript. B.C.C.: Project PI, acquired funding, collected samples, supervised research, and revised the manuscript. J.C.P.: Designed the

study, was Chief Scientist, assisted with sample collection, and revised the manuscript. O.S.K.: Project PI, designed the study, acquired funding, supervised research, and revised the manuscript.

## Competing interests

The authors declare no competing interests.
