## [Transparent Peer Review file · Nature Communications]

Genetic isolation and metabolic complexity of an Antarctic subglacial microbiome

Corresponding Author: Dr Ok-Sun Kim

Version 0:

Reviewer comments:

Reviewer #1

(Remarks to the Author)

The authors have addressed my concerns, although more detail (line numbers and text) of changes would have been helpful and speeded up this review.

Reviewer #3

(Remarks to the Author)

The author has successfully addressed many of my technical concerns regarding their work. However, I still find the metabolic functions inferred from the incomplete Metagenome-Assembled Genomes (MAGs) to be questionable. The main strengths of this study lie in its novel location and the methodology employed in single-cell genomics.

I was expecting to see more results pertaining to the unique functions that underpin the microbial community within this subglacial lake. Insights into these functions are crucial, as they could shed light on the survival mechanisms in extraterrestrial environments.

Reviewer #4

(Remarks to the Author)

Hwang et. al present an in-depth study of microbial communities inhabiting the subglacial ecosystem beneath the West Antarctic Ice Sheet, focusing on how these microbes survive and evolve under energy-limited conditions. By obtaining samples from the Mercer Subglacial Lake, the authors successfully recovered 1,374 single-cell amplified genomes, which holds significant scientific value in enhancing our understanding of microbial diversity and metabolic mechanisms in this extreme environment. While the study presents significant findings, several major concerns need to be addressed before it can be considered for publication.

Major comments:

Insufficient Evidence for Evolutionary Claims:

The current analysis primarily reveals genomic isolation from a phylogenetic perspective without adequately comparing functional characteristics with other environmental genomes. Therefore, caution is urged in making broad claims regarding microbial evolution based solely on these observations. Another question is the split between Result 3 and Result 2. The manuscript would benefit from incorporating additional functional comparative analyses with other relevant environments to substantiate evolutionary claims.

In-depth Analysis of Habitat:

While the classification of samples into water column and sediment is helpful, the authors introduce a category termed "lake_small_SAG" based on sample fraction (<0.2 µm), but fail to discuss this portion adequately. It is crucial to explore whether these organisms may represent parasitic or symbiotic species (e.g. p__Patescibacteria). Additionally, the inclusion

of SLW samples in the 16S analysis requires clarification regarding its rationale.

Relationship Between Species and Metabolic Functions:

There appears to be a disconnect between certain species and their expected metabolic functions. For example, the gene *nxrA* was detected in only a few SAGs of the genus *Nitrotoga*, which may be attributed to issues related to genome completeness; this could be further validated with the NCyC database [1]. Additionally, the genomic potential of the Calvin-Benson-Bassham cycle under anoxic conditions necessitates more thorough discussion. For instance, the mixotrophic genus *Polaromonas* has demonstrated its ability to facilitate carbon fixation in groundwater environments [2], yet its carbon fixation capabilities on glacier surfaces have received minimal attention. Therefore, it is crucial to investigate the differences in the CBB cycle between subglacial, supraglacial, and similar environments. Such insights could enhance our understanding of primary production in the SLM.

1. <https://doi.org/10.1093/bioinformatics/bty741>

2. 10.1038/s41396-021-01163-x

Concerns Regarding methodology :

Given that previous reviews raised concerns about the quality of single-cell genomes, and acknowledging the significant efforts made by the authors to enhance data confidence, I do not intend to comment on the quality of the SAG. However, the analysis related to ASV (Amplicon Sequence Variants) appears to lack depth and significance. If there is a strong match between the 16S sequence in SAGs and the ASV sequence, the relative abundance of SAGs in the community can be inferred from this relationship, providing insight into whether SAGs belong to rare species.

Additionally, the authors mention a mismatch between ASV species and SAG data, which may be attributed to discrepancies in database classification systems. It would be beneficial to utilize the NCBI 16S database for ASV taxonomy annotation, as there is a correspondence between GTDB classification and NCBI classification.

Minor comments :

Line 80-83: It is recommended to add a sampling map and a diagram of the environmental parameters to facilitate readers' understanding of the sample and habitat characteristics.

Line 95-97: No result or discussion shows the strain heterogeneity in the manuscript.

Line 127-134: It is suggested to add a bar chart illustrating taxonomy composition and a comparison of genome sizes among different groups. The lake_small_SAG may represent species that have adapted to their environment through genomic streamlining.

Line 147: Why is the beta diversity analysis weighted by abundance when SAGs do not correspond to actual abundance information?

Line 204-206: The genomes within the lake_small_SAG group may have potential ecological significance, such as more amino acid deficiencies and reliance on public production for survival.

Line 208-210: The essence of pan-genome analysis lies in examining differences across individual genomes to capture core and accessory genes. The structural variations among different genomes could serve as evidence for evolutionary processes.

Line 243: There is a missing space in "Thep__Actinomycetota."

Line 339-340: Why is the oxygen condition of SLW mentioned?

Line 367-369: Denitrification processes typically occur under anoxic conditions. How can the presence of denitrification genes in the water column be explained?

Version 1:

Reviewer comments:

Reviewer #3

(Remarks to the Author)

All my concerns have been fully addressed

(Remarks on code availability)

The code have been fully disclosed in the github

Reviewer #4

(Remarks to the Author)

The authors have adequately addressed most concerns. Only minor taxonomy formatting suggestions remain: Alphanumeric placeholder names (e.g., GW2011-AR1 family) should not be italicized; Taxonomic rank prefixes (such as 'p__' or 'f__') should be omitted for cleaner presentation.

(Remarks on code availability)

Genetic isolation and metabolic complexity of an Antarctic subglacial microbiome

Kyung Mo Kim, Kyuin Hwang, Hanbyul Lee, Ahnna Cho, Christina L. Davis, Brent C. Christner, John C. Priscu, and Ok-Sun Kim

REVIEWER COMMENTS

Reviewer #1 (Remarks to the Author):

The authors have addressed my concerns, although more detail (line numbers and text) of changes would have been helpful and speeded up this review.

Response: Thank you. Your previous comments were very helpful to improve our manuscript.

Reviewer #3 (Remarks to the Author):

The author has successfully addressed many of my technical concerns regarding their work. However, I still find the metabolic functions inferred from the incomplete Metagenome-Assembled Genomes (MAGs) to be questionable. The main strengths of this study lie in its novel location and the methodology employed in single-cell genomics.

Response: Thank you for this comment. To evaluate whether the low completeness of our SAGs (38% on average) affects the quantitative estimation of metabolic potential, we tested whether the genomic proportions of metabolic pathways change significantly when high-quality genomes are artificially reduced. For this analysis, we selected 1,229 GTDB genomes (average completeness of 82.4%) that are the closest genetic matches to our SAGs and reduced them to ~38% completeness to match our SAGs. This genome reduction process was randomly repeated 1,000 times using the Monte Carlo permutation test to generate distributions of the genomic proportions for each of the 70 metabolic pathways. Statistical comparison between the distributions from reduced genomes and the true genomic proportions from the original GTDB genomes revealed that only 4 pathways were significantly underestimated (Supplementary Table 9), indicating that approximately 94% of the pathways (66 out of 70) are not significantly underestimated by genome incompleteness at this level. In summary, our artificial genome reduction experiment suggests that a genome completeness of ~38% does not significantly underestimate the genomic proportions for most metabolic pathways. Furthermore, limiting the analysis to only higher-quality SAGs can bias community structure. For instance, the relative abundance of *p__Actinomycetota* increased disproportionately from 45.6% to 74.5% when SAGs with <50% completeness were excluded. This filtering also removed rare but ecologically relevant taxa, such as *Nitrosarchaeum* (completeness <22.8%) and ammonia-oxidizing Nitrosomonadaceae (48.2% completeness). Therefore, we included all 1,374 SAGs in the

functional analysis. Further details are provided in Results (lines 247-286) and Materials and Methods (lines 946-961) of the revised manuscript.

I was expecting to see more results pertaining to the unique functions that underpin the microbial community within this subglacial lake. Insights into these functions are crucial, as they could shed light on the survival mechanisms in extraterrestrial environments.

Response: Thank you for this comment. To explore the functional uniqueness of the subglacial lake microbial ecosystem relative to surface microbial communities, we initially considered comparing SLM SAG genomes with publicly available environmental genomic samples. However, despite searching NCBI and IMG databases for environmental samples that have undergone random single-cell sorting followed by sufficient SAG sequencing, we found very few available datasets (e.g., groundwater SAGs from NCBI Bioprojects PRJNA627556 and from PRJNA362739). Although many shotgun metagenome data sequenced from terrestrial freshwater environments are available, shotgun metagenomics and single-cell genomics are inherently biased in different ways (e.g., biases from PCR amplification in shotgun metagenomics versus biases from cell sorting and whole-genome amplification in single-cell genomics), making direct comparison of sequence data from these two approaches inappropriate¹. Therefore, we instead investigated metabolic pathways that are enriched or depleted in SLM taxa compared to microbes isolated from other environments (e.g., ice, groundwater, soil, acid-mine drainage, and wastewater) within the phylogenomic tree framework. Specifically, we compared the genomic proportions of 79 metabolic pathways between SLM taxa (71% of 1,374 SAGs) and their closest evolutionary relatives (sister taxa; 1,196 GTDB genomes) across the 18 most abundant genera in SLM. This analysis revealed that no metabolic pathways were consistently depleted or enriched across all the genera (Supplementary Table 10), suggesting that our SAG dataset does not include metabolic pathways unique to the Antarctic subglacial lake. To further explore unique metabolic functions in SLM, future studies will need to experimentally characterize the large number of proteins lacking functional annotations (i.e., <30% sequence similarity to any known proteins²; Fig. 3c). These uncharacterized proteins likely harbor novel functions that may play important roles in microbial adaptation to this extreme ecosystem. No changes have been made to the revised manuscript regarding this issue.

Reviewer #4 (Remarks to the Author):

Hwang et. al present an in-depth study of microbial communities inhabiting the subglacial ecosystem beneath the West Antarctic Ice Sheet, focusing on how these microbes survive and evolve under energy-limited conditions. By obtaining samples from the Mercer Subglacial Lake, the authors successfully recovered 1,374 single-cell amplified genomes, which holds significant scientific value in enhancing our understanding of microbial diversity and metabolic mechanisms in this extreme environment. While the study presents significant findings, several major concerns need to be addressed before it can be considered for publication.

Response: Thank you for this evaluation.

Major comments:

Insufficient Evidence for Evolutionary Claims:

The current analysis primarily reveals genomic isolation from a phylogenetic perspective without adequately comparing functional characteristics with other environmental genomes. Therefore, caution is urged in making broad claims regarding microbial evolution based solely on these observations. Another question is the split between Result 3 and Result 2. The manuscript would benefit from incorporating additional functional comparative analyses with other relevant environments to substantiate evolutionary claims.

Response: Thank you for this comment. In order to examine whether SLM-inhabiting microbes have functionally diverged from microbes in other environments, we compared the relative abundance (i.e., the proportion of genomes) of metabolic pathways between SLM taxa and their sister taxa within the phylogenomic trees of the most abundant genera (covering 71% of the 1,374 SAGs). Notably, the sister taxa originate from a wide range of environmental sources (e.g., ice, groundwater, soil, acid mine drainage, and wastewater) and represent the closest evolutionary relatives of the SLM SAGs. This makes them ideal comparators for testing the functional divergence of SLM microbes from other microbiomes. Our results demonstrate clear functional divergence between SLM-inhabiting microbes and their phylogenetic relatives (Supplementary Table 10). Consequently, both the phylogenomic and functional divergence between SLM and sister taxa support the genetic isolation of Antarctic subglacial microbes from surface biomes. To illustrate this, we incorporated graphs showing metabolic pathway enrichments into each phylogenomic tree of the most abundant genera (Supplementary Figs. 7–25). Further details are provided in Results (lines 216-220), Discussion (458-461), and Materials and Methods (901-903) of the revised text.

In-depth Analysis of Habitat:

While the classification of samples into water column and sediment is helpful, the authors introduce a category termed "lake_small_SAG" based on sample fraction ($<0.2 \mu\text{m}$), but fail to discuss this portion adequately. It is crucial to explore whether these organisms may represent parasitic or symbiotic species (e.g. p__Patescibacteria). Additionally, the inclusion of SLW samples in the 16S analysis requires clarification regarding its rationale.

Response: Thank you for your comments. We have added more detailed descriptions of genome reduction and limited metabolic capability exhibited by lake_small_SAGs in Results (lines 381-397) of the revised text. Based on KEGG functional annotations of 202 Patescibacteria SAGs, we further discussed genome reduction, lack of biosynthetic pathways, energy conservation strategies, and possible lifestyles of these ultrasmall bacteria in Results (lines 398-439) and Discussion (lines 593-632) of the revised text. As shown in Supplementary Fig. 2b, there is no $<0.2 \mu\text{m}$ sample available for SLW. In fact, the comparison of 16S rRNA sequences between SLW ASVs and SLM SAGs was conducted to (1) evaluate how similar the

community structure and taxonomic composition of SLW are to those of SLM (Supplementary Fig. 2b-c and Supplementary Fig. 4), and to (2) assess why taxa abundant in SLW (e.g., *Thiobacillus* and *Sideroxydans*) were not detected in our SLM SAG dataset (Supplementary Fig. 5). These points were addressed in Results (lines 165-171) and Supplementary Text (lines 54-85) of the revised manuscript.

Relationship Between Species and Metabolic Functions:

There appears to be a disconnect between certain species and their expected metabolic functions. For example, the gene *nxrA* was detected in only a few SAGs of the genus *Nitrotoga*, which may be attributed to issues related to genome completeness; this could be further validated with the NCyC database [1]. Additionally, the genomic potential of the Calvin-Benson-Bassham cycle under anoxic conditions necessitates more thorough discussion. For instance, the mixotrophic genus *Polaromonas* has demonstrated its ability to facilitate carbon fixation in groundwater environments [2], yet its carbon fixation capabilities on glacier surfaces have received minimal attention. Therefore, it is crucial to investigate the differences in the CBB cycle between subglacial, supraglacial, and similar environments. Such insights could enhance our understanding of primary production in the SLM.

1. <https://doi.org/10.1093/bioinformatics/bty741>

2. 10.1038/s41396-021-01163-x

Response: Thank you for your comments. Upon comparison against the NCycDB database, the three putative *nxrA* sequences found among the 108 *g__Nitrotoga* SAGs were identified as nitrate reductase (*narG*), suggesting that none of the *g__Nitrotoga* SAGs exhibit the metabolic potential for nitrite oxidation. To assess the broader presence of nitrite oxidizers in SLM, we screened all protein sequences from the 1,374 SAGs against NCycDB. This analysis identified only one *nxrA* gene in the SAG SLM_LV1_0208-B14 (*f__Opitutaceae* in *p__Verrucomicrobiota*), indicating the possible presence of nitrite-oxidizing bacteria in the SLM environment. Further discussion has been included in lines 541-562 of the revised text. In response to your suggestion, we also discussed primary production in SLM in comparison to other similar environments, linking metabolic functions with taxonomic information. From this, we found that carbon fixation may occur along oxygen gradient from oxic to anoxic conditions, with sulfur-oxidizing chemolithoautotrophs becoming important in chemosynthesis as oxygen availability declines. Notably, *g__Nitrotoga* may play a critical role in primary production under both suboxic and anoxic conditions. Additionally, SAGs assigned to *g__Polaromonas* appear to be mixotrophic, possessing both heterotrophic and autotrophic capabilities. These traits seem are likely consistent with *Polaromonas* strains isolated from groundwater environments³, but differ from supraglacial *Polaromonas*, which have not been shown to fix carbon dioxide⁴. More details are provided in Discussion (lines 563-592) of the revised text.

Concerns Regarding methodology :

Given that previous reviews raised concerns about the quality of single-cell genomes, and acknowledging the significant efforts made by the authors to enhance data confidence, I do not intend to comment on the quality of the SAG. However, the analysis related to ASV (Amplicon

Sequence Variants) appears to lack depth and significance. If there is a strong match between the 16S sequence in SAGs and the ASV sequence, the relative abundance of SAGs in the community can be inferred from this relationship, providing insight into whether SAGs belong to rare species.

Response: In response to your comment, we evaluated the feasibility of mapping ASV abundance onto SAGs by assessing their rank correlation. ASVs from the SLM dataset⁵ were clustered into OTUs using VSEARCH v2.24 at >97% sequence similarity. We then identified corresponding SAGs by BLASTn-searching 6,133 SLM ASVs against the 16S rRNA gene sequences of the SAGs. As a result, the relative abundance of SAGs (x-axis in the figure below) did not correlate with the relative OTU abundance of their corresponding ASVs (y-axis) in either sediments or the water column ($R^2 < 0.1$). This analysis revealed a discrepancy, likely due to uneven 16S primer affinity across taxa⁶ and biases inherent to single-cell sorting (e.g., cell size, lysis efficiency, particle association)⁷, suggesting that ASV abundance may not reliably represent SAG abundance. In fact, microbial strains within a species can often share 100% identical 16S rRNA sequences, meaning that distinct SAGs can correspond to the same ASV. As a result, ASV-based analyses may obscure the ecological distinctiveness of individual strains that SAGs are capable of resolving (Supplementary Fig. 28). Furthermore, among the 1,374 SAGs, only 686 contained 16S rRNA sequences, while the remaining SAGs lacked them due to low genome completeness. Consequently, given the limited representation of SAGs with 16S data (668 of 1,374) and the inability of ASVs to resolve microbial strain-level differences, ASV abundance does not serve as a reliable proxy for SAG abundance. Thus, we chose not to include the analysis of mapping ASV abundance onto SAGs in the revised manuscript.

Additionally, the authors mention a mismatch between ASV species and SAG data, which may be attributed to discrepancies in database classification systems. It would be beneficial to utilize the NCBI 16S database for ASV taxonomy annotation, as there is a correspondence between GTDB classification and NCBI classification.

Response: Thank you for your insightful suggestion. The taxa of interest regarding the observed mismatch were *Sideroxydans* and *Thiobacillus*, which were abundant in the ASV dataset from SLM⁵ but not detected in our SAGs. Notably, the taxonomic designations of these genera are consistent across different classification systems: GTDB refers to them as *g__Sideroxyarcus* (a synonym of *g__Sideroxydans*) and *g__Thiobacillus*; SILVA designates them as *Sideroxydans* and *Thiobacillus*; and NCBI uses *g__Sideroxydans* and *g__Thiobacillus*. Due to this consistency, it is unlikely that discrepancies among classification systems account for the mismatch between the ASV and SAG data. Instead, we re-evaluated whether 16S rRNA sequences from SAGs cluster with ASVs by reconstructing phylogenetic trees using deeper sequence sampling. For *Sideroxydans*, we generated a phylogeny that included: (1) SAGs classified within GTDB as *f__Gallionellaceae* (64 16S rRNA sequences), (2) GTDB representative genomes from *f__Gallionellaceae* (68 sequences), (3) ASVs assigned to *Gallionellaceae* (143 sequences), and (4) SILVA sequences annotated as *Sideroxydans* (332 sequences). This analysis revealed that a GTDB-designated taxon *g__39-52-13* clustered with *Sideroxydans* sequences in SILVA (accession= EU030485). This cluster also included seven ASVs from the SLM dataset, each exhibiting >98.4% sequence similarity to *Sideroxydans* sequences in SILVA (Supplementary Fig. 5). Based on this phylogenetic evidence, we concluded that *g__39-52-13* (*f__Gallionellaceae*) in GTDB corresponds to *Sideroxydans* (*f__Gallionellaceae*), and that this genus is present in both the single-cell and ASV datasets. Further details are provided in lines 66-85 of Supplementary Text. In the case of *Thiobacillus*, none of the 16S rRNA sequences from SAGs matched any of the *Thiobacillus* ASVs from the SLM dataset. Furthermore, none of the 1,374 SAGs were assigned to the family *f__Thiobacillaceae* in the GTDB taxonomy. Given this clear absence of *Thiobacillus* among our SAGs, we did not perform further analysis through phylogenetic reconstruction for this genus.

Minor comments :

Line 80-83: It is recommended to add a sampling map and a diagram of the environmental parameters to facilitate readers' understanding of the sample and habitat characteristics.

Response: now included as Fig. 1

Line 95-97: No result or discussion shows the strain heterogeneity in the manuscript.

Response: We have added sentences addressing strain heterogeneity in Results (lines 238-246) and Supplementary Text (lines 97-119) of the revised manuscript.

Line 127-134: It is suggested to add a bar chart illustrating taxonomy composition and a comparison of genome sizes among different groups. The lake_small_SAG may represent species that have adapted to their environment through genomic streamlining.

Response: We have added the requested bar chart as Supplementary Fig. 32. We reported the genome sizes of lake_small_SAGs and Patescibacteria in Results (lines 381-397 and 398-439) and discussed the survival advantage of subglacial microbes conferred by genome streamlining in oligotrophic environments in Discussion (lines 593-632) of the revised text.

Line 147: Why is the beta diversity analysis weighted by abundance when SAGs do not correspond to actual abundance information?

Response: As shown in the scatter plots above, there was no correlation between SAG-derived 16S rRNA sequences and metabarcoding-derived 16S rRNA sequences in taxon abundance. This incongruence is not unexpected, as a similar finding has been reported recently¹. Technical biases inherent to single-cell genomics (e.g., cell size and particle association in cell sorting, and GC content during whole genome amplification) can skew the observed abundance of SAGs. In contrast, metabarcoding-based 16S rRNA surveys (e.g., SLW's ASVs shown in Supplementary Fig. 2b–c) are affected by biases in cell lysis efficiency and primer affinity, both of which influence OTU abundance. As such, metabarcoding-based 16S rRNA sequences do not necessarily reflect the true abundance of microbial taxa in environmental samples. Despite these limitations, they have been widely used in weighted beta-diversity analyses, supporting the use of similarly biased SAG-based abundance data in such contexts. A direct comparison of bias magnitude between these two approaches would be valuable but is beyond the scope of this study.

Line 204-206: The genomes within the lake_small_SAG group may have potential ecological significance, such as more amino acid deficiencies and reliance on public production for survival.

Response: Thank you for this insightful comment. We have added our description of the metabolic potential of lake_small_SAGs, emphasizing their metabolic deficiencies in carbon, nitrogen, sulfur, and iron cycling. In particular, we included new details on the ecophysiology of Patescibacteria (the majority of lake_small_SAGs), which lack biosynthetic pathways for amino acids, fatty acids, and nucleotides (Fig. 6). Our analysis suggests that Patescibacteria are likely fermentative organisms, as genes associated with oxidative phosphorylation and the TCA cycle were not detected in their SAGs. These results indicate that they may contribute to carbon cycling via fermentation in SLM. Moreover, Patescibacteria appear to rely on the external uptake of sugars for energy conservation and may engage in epi-symbiotic interactions with host bacteria to acquire essential nutrients (e.g., DNA and peptides). We have incorporated these details into Results (lines 381-439) and further discussed their survival strategies, lifestyles, and ecological significance in Discussion (lines 593-632) of the revised text.

Line 208-210: The essence of pan-genome analysis lies in examining differences across individual genomes to capture core and accessory genes. The structural variations among different genomes could serve as evidence for evolutionary processes.

Response: Thank you for pointing out our misunderstanding regarding the use of the term *pan-genome*. We have removed it throughout the revised manuscript. We agree that examining structural variations among SAGs can provide valuable insights into the evolutionary dynamics of subglacial microbes. However, due to the fragmented nature and relatively low completeness of our SAGs, it is challenging to reliably resolve patterns of structural variation. Accordingly, we did not perform such analyses in this study.

Line 243: There is a missing space in “Thep__Actinomycetota.”

Response: corrected

Line 339-340: Why is the oxygen condition of SLW mentioned?

Response: The sentence was removed as the oxygen concentration in SLW is not necessary for the discussion.

Line 367-369: Denitrification processes typically occur under anoxic conditions. How can the presence of denitrification genes in the water column be explained?

Response: This may be due to anoxic microenvironments or because SLM occasionally becomes anoxic when oxygen is depleted by chemical oxidation and microbial respiration. These explanations have been added to Discussion (lines 523-528) of the revised text.

References

- 1 Goordial, J. *et al.* Microbial diversity and function in shallow subsurface sediment and oceanic lithosphere of the Atlantis Massif. *MBio* **12**, 10.1128/mbio.00490-00421 (2021).
- 2 Murzin, A. G., Brenner, S. E., Hubbard, T. & Chothia, C. SCOP: a structural classification of proteins database for the investigation of sequences and structures. *Journal of molecular biology* **247**, 536-540 (1995).
- 3 Taubert, M. *et al.* Bolstering fitness via CO₂ fixation and organic carbon uptake: mixotrophs in modern groundwater. *The ISME journal* **16**, 1153-1162 (2022).
- 4 Franzetti, A. *et al.* Light-dependent microbial metabolisms drive carbon fluxes on glacier surfaces. *The ISME journal* **10**, 2984-2988 (2016).
- 5 Davis, C. L. *et al.* Biogeochemical and historical drivers of microbial community composition and structure in sediments from Mercer Subglacial Lake, West Antarctica. *ISME communications* **3**, 8 (2023).

- 6 Han, D. *et al.* Multicenter assessment of microbial community profiling using 16S rRNA gene sequencing and shotgun metagenomic sequencing. *Journal of advanced research* **26**, 111-121 (2020).
- 7 Woyke, T., Doud, D. F. R. & Schulz, F. The trajectory of microbial single-cell sequencing. *Nature Methods* **14**, 1045-1054 (2017). <https://doi.org/10.1038/nmeth.4469>

Genetic isolation and metabolic complexity of an Antarctic subglacial microbiome

Kyung Mo Kim, Kyuin Hwang, Hanbyul Lee, Ahnna Cho, Christina L. Davis, Brent C. Christner, John C. Priscu, and Ok-Sun Kim

REVIEWER COMMENTS

Reviewer #3 (Remarks to the Author):

All my concerns have been fully addressed

Reviewer #3 (Remarks on code availability):

The code have been fully disclosed in the github

Reviewer #4 (Remarks to the Author):

The authors have adequately addressed most concerns. Only minor taxonomy formatting suggestions remain: Alphanumeric placeholder names (e.g., GW2011-AR1 family) should not be italicized;

Response: Thank you. Alphanumeric placeholder names are now non-italicized throughout the revised manuscript (e.g., GW2011-AR1 in line 133, UBA1004 in line 136, etc.).

Taxonomic rank prefixes (such as 'p__' or 'f__') should be omitted for cleaner presentation.

Response: Thank you. The taxonomic rank prefixes have been removed throughout the revised manuscript (e.g., Actinomycetota in line 127, *Nitrosarchaeum* in line 133, etc.).